# Pseudo-Likelihood Inference

**Theo Gruner** [1,2]    **Boris Belousov** [3]    **Fabio Muratore** [4]
**Daniel Palenicek** [1,2]    **Jan Peters** [1,2,3,5]
[1] Intelligent Autonomous Systems Group, Technical University of Darmstadt
[2] hessian.AI    [3] German Research Center for AI (DFKI)
[4] Bosch Center for Artificial Intelligence    [5] Centre for Cognitive Science
theo_sunao.gruner@tu-darmstadt.de

## Abstract

Simulation-Based Inference (SBI) is a common name for an emerging family of approaches that infer the model parameters when the likelihood is intractable. Existing SBI methods either approximate the likelihood, such as Approximate Bayesian Computation (ABC), or directly model the posterior, such as Sequential Neural Posterior Estimation (SNPE). While ABC is efficient on low-dimensional problems, on higher-dimensional tasks, it is generally outperformed by SNPE which leverages function approximation. In this paper, we propose Pseudo-Likelihood Inference (PLI), a new method that brings neural approximation into ABC, making it competitive on challenging Bayesian system identification tasks. By utilizing integral probability metrics, we introduce a smooth likelihood kernel with an adaptive bandwidth that is updated based on information-theoretic trust regions. Thanks to this formulation, our method (i) allows for optimizing neural posteriors via gradient descent, (ii) does not rely on summary statistics, and (iii) enables multiple observations as input. In comparison to SNPE, it leads to improved performance when more data is available. The effectiveness of PLI is evaluated on four classical SBI benchmark tasks and on a highly dynamic physical system, showing particular advantages on stochastic simulations and multi-modal posterior landscapes.

## 1   Introduction

Parametric stochastic simulators are a well-established tool for predicting the behavior of real-world phenomena. These statistical models find widespread use in various scientific fields such as physics, economics, biology, ecology, computer science, and robotics, where they help to gain knowledge about the underlying stochastic processes [27, 34] or generate additional data for subsequent downstream tasks [38]. In both cases, the practitioner seeks to explain the observations as accurately as possible while incorporating all available information. The output of such a simulator is largely determined by its parameters and their values. When estimating these parameters using Bayesian inference, given observations from a physical system, which are inevitably subject to measurement noise, we obtain a distribution over values instead of a point estimate. Additionally, there might be several parameter configurations yielding the same observation, hence rendering the resulting distribution to be multi-modal. Moreover, the likelihood function might be unknown or too expensive to evaluate for many practical use cases. The combination of these difficulties makes obtaining a posterior distribution over simulator parameters challenging for state-of-the-art inference methods, both regarding effectiveness and efficiency.

SBI approaches address the issue of intractable likelihoods by using (stochastic) simulators as forward models to generate observations from proposal distributions over parameters. The approaches are also often called *likelihood-free*, which can be easily misunderstood since some of them directly approximate the likelihood [7]. ABC is a family of SBI methods that approximate the posterior with

37th Conference on Neural Information Processing Systems (NeurIPS 2023).

a set of weighted particles which are obtained from Monte Carlo simulations and updated based on an empirical estimation of the intractable likelihood [48]. For an ABC approach to work well, three criteria have to be fulfilled: (i) the likelihood kernel is capable of measuring the similarity of observations meaningfully, (ii) the proposal distribution samples close to the posterior, (iii) the decision-making rule balances between accepting a sufficient amount of samples from the proposal and steering inference towards the posterior distribution. Constructing a suitable likelihood kernel often means tailoring summary statistics to the problem at hand. However, recent advances promise to replace this heuristic-based process by employing Integral Probability Metrics (IPMs) to measure statistical distances in observation space [3, 16, 17]. While these methods significantly increase the required number of simulations, approximations of the statistical distances [25, 8] can be computed in parallel, hence facilitating the parallelization of the whole inference pipeline. Following up on the shortcomings of ABC, the family of SNPE approaches provide Bayesian inference methods that leverage conditional neural density estimators to approximate the posterior [40, 24, 41, 19]. The benchmarking study of Lueckmann et al. [32] concludes that, generally, SNPE approaches are to be preferred over ABC as they are superior in terms of expressibility and accuracy across a wide range of benchmarking tasks. However, it is important to point out that the analysis of the posterior inference has solely been reported for single observations. These single-sample scenarios favor SNPE in high dimensions since ABC relies on summary statistics to evaluate the likelihood. Therefore, it remains an open question whether SNPE methods can transfer their benefits to settings where the (approximated) posterior is conditioned on multiple observations at once.

**Contributions.**  By deriving the ABC posterior from a constrained variational inference objective, we introduce a novel SBI method called Pseudo-Likelihood Inference (PLI). Our formulation allows for approximating the posterior with neural density estimators. PLI updates this posterior from pseudo-likelihoods which are exponentially transformed statistical distances computed using IPMs. To further remove heuristics from the inference process, we derive an adaptive bandwidth update of PLI's likelihood kernel that bounds the loss of information based on information-geometric trust-region principles. This way, PLI can update its neural posterior solely given observations from a (stochastic) black-box simulator. Moreover, the usage of IPMs enables PLI to simultaneously condition on a variable number of observations, while SNPE methods need to concatenate them and, therefore, degrade when the number of observations increases. We compare PLI against ABC and one SNPE method on two SBI benchmarking tasks as well as a highly dynamical double pendulum task. For both, ABC and PLI, we investigate two IPMs: the Maximum Mean Discrepancy (MMD) and the Wasserstein distance. Motivated by recent benchmarking results, we chose Automatic Posterior Transformation (APT) [24] to represent SNPE approaches. Our experiments investigate the dependency of the trained density estimator's performance on the number of observations, where performance is measured in observation as well as in parameter space. We show the merits and disadvantages of all methods and conclude with concrete recommendations.

## 2  Bayesian inference with intractable likelihoods

The objective of Bayesian inference is to find the posterior parameter distribution $p(\boldsymbol{\xi}|\mathbf{x}_{1:N}^\star)$ given a set of reference data points $\mathbf{x}_{1:N}^\star$ which are assumed to be drawn from the likelihood model $p(\mathbf{x}|\boldsymbol{\xi})$. Given a prior belief over the parameters $p(\boldsymbol{\xi})$, the posterior is expressed via Bayes' rule

$$p(\boldsymbol{\xi}|\mathbf{x}_{1:N}^\star) \propto p(\mathbf{x}_{1:N}^\star|\boldsymbol{\xi})\, p(\boldsymbol{\xi}). \tag{1}$$

In the following, we describe SBI methodologies that aim to approximate the posterior (1) when the likelihood is given by a simulator model, from which only sampling is possible, $\mathbf{x}_{1:M} \sim p(\mathbf{x}|\boldsymbol{\xi})$ but evaluating the likelihood is infeasible.

### 2.1  Approximate Bayesian computation

ABC methods perform Bayesian inference without explicitly computing the likelihood function $p(\mathbf{x}_{1:N}^\star|\boldsymbol{\xi}) = \int p(\mathbf{x}_{1:N}^\star|\mathbf{x}_{1:M}, \boldsymbol{\xi})p(\mathbf{x}_{1:M}|\boldsymbol{\xi})\, \mathrm{d}\mathbf{x}_{1:M}$ [48]. Instead, they approximate it

$$\tilde{p}_\beta(\mathbf{x}_{1:N}^\star|\boldsymbol{\xi}) \propto \int K_\beta(D(\mathbf{x}_{1:N}^\star, \mathbf{x}_{1:M}))\, p(\mathbf{x}_{1:M}|\boldsymbol{\xi})\mathrm{d}\mathbf{x}_{1:M} \tag{2}$$

using Monte Carlo samples $\mathbf{x}_{1:M}$ from the simulator as the reference points and smoothening them with a kernel $K_\beta(D(\mathbf{x}_{1:N}^\star, \mathbf{x}_{1:M}))$ [28]. The kernel assesses the similarity of the reference data

$\mathbf{x}_{1:N}^{\star}$ and the simulated data $\mathbf{x}_{1:M}$ based on a distance measure $D$, the kernel type $K$, and the kernel bandwidth $\beta$. The uniform kernel $\mathbb{1}_{\{D(\mathbf{x}_{1:N}^{\star}, \mathbf{x}_{1:M}) \leq \beta\}}$ (see Table A.1) has emerged as the default kernel choice of many ABC methods [49, 15, 30, 31]. In this case, the bandwidth represents a rejection threshold that assigns zero probability to all parameters whose simulations lie outside of the $\beta$-ball in terms of the distance $D$. The uniform kernel exhibits the favorable characteristic of converging to the likelihood in the limit [28]

$$\lim_{\beta \to 0} \tilde{p}_{\beta}(\mathbf{x}_{1:N}^{\star}|\boldsymbol{\xi}) = \int \mathbb{1}_{\{\mathbf{x}_{1:N}^{\star}\}}(\mathbf{x}_{1:M}) p(\mathbf{x}_{1:M}|\boldsymbol{\xi}) \mathrm{d}\mathbf{x}_{1:M} = p(\mathbf{x}_{1:N}^{\star}|\boldsymbol{\xi}). \tag{3}$$

Once the approximate likelihood (2) is obtained, ABC draws samples from the approximate posterior

$$\tilde{p}_{\beta}(\boldsymbol{\xi}|\mathbf{x}_{1:N}^{\star}) \propto \tilde{p}_{\beta}(\mathbf{x}_{1:N}^{\star}|\boldsymbol{\xi}) \, p(\boldsymbol{\xi}). \tag{4}$$

There exist multiple ways of implementing this sampling procedure. In rejection ABC [49], the simplest form, proposal parameters are drawn from the prior distribution $p(\boldsymbol{\xi})$ and are accepted if the simulated data falls close to the true data, as measured by the kernel function. While rejection ABC yields a simple algorithm with desirable convergence properties, finding posterior samples for small bandwidths $\beta$ in high dimensions often becomes computationally infeasible [33]. Therefore, the research on ABC focuses on three directions of improvement: (i) replacing the prior with a sequentially updated proposal distribution $\pi_t(\boldsymbol{\xi})$ to reduce the search space during sampling, (ii) adapting the bandwidth $\beta$ to draw samples with an appropriate acceptance rate, and (iii) finding sufficient statistics to represent the simulated output in low dimensions [48]. MCMC-ABC [33] and SMC-ABC [47, 51, 15, 31] build upon sampling strategies based on Markov Chain Monte Carlo (MCMC) and Sequential Monte Carlo (SMC) to sequentially update the proposal distribution. MCMC-ABC does not allow for an adaptive bandwidth, and thus, SMC sampling strategies have evolved as the leading ABC methods for these cases [15].

## 2.2 Sequential Monte Carlo ABC

SMC–ABC builds on SMC samplers introduced by Del Moral et al. [14]. Fundamentally, SMC-ABC approximates the posterior distribution through a sequence of intermediate *target posterior* distributions $\tilde{p}_{\beta_t}(\boldsymbol{\xi}|\mathbf{x}_{1:N}^{\star})$ (4) that are characterized by an adaptable bandwidth parameter $\beta_t$, where $t$ denotes the inference time. Furthermore, SMC-ABC uses importance sampling from a sequentially updated proposal distribution $\pi_t(\boldsymbol{\xi})$ to improve the sample efficiency. The proposal distribution is represented by an empirical distribution $\pi_t(\boldsymbol{\xi}) = 1/M \sum_{i=0}^{M} \delta_{\boldsymbol{\xi}_t^{(i)}}(\boldsymbol{\xi})$ that is defined through a set of particles $\{\boldsymbol{\xi}_t^{(i)}\}$. Importance sampling then enables the approximation of the target posterior $\tilde{p}_{\beta_t}(\boldsymbol{\xi}|\mathbf{x}_{1:N}^{\star})$ from the proposal distribution

$$\tilde{p}_{\beta_t}(\boldsymbol{\xi}|\mathbf{x}_{1:N}^{\star}) \approx q_t(\boldsymbol{\xi}) = \sum_{i=1}^{M} W_t^{(i)} \delta_{\boldsymbol{\xi}_t^{(i)}}(\boldsymbol{\xi}); \quad W_t^{(i)} = \frac{\tilde{p}_{\beta_t}(\boldsymbol{\xi}_t^{(i)}|\mathbf{x}_{1:N}^{\star})}{\pi_t(\boldsymbol{\xi}^{(i)})}. \tag{5}$$

Here, $W_t$ are the weights between the target posterior and the proposal distribution. SMC-ABC methods follow three steps to carry out inference for the next target posterior $\tilde{p}_{\beta_{t+1}}(\boldsymbol{\xi}|\mathbf{x}_{1:N}^{\star})$: (i) A new bandwidth $\beta_{t+1}$ of the target posterior $\tilde{p}_{\beta_{t+1}}(\mathbf{x}_{1:N}^{\star}|\boldsymbol{\xi})$ is estimated. Typically, the update is based on heuristics, such as the Effective Sample Size (ESS) [15] to ensure that the particle variance does not degrade. (ii) New proposal particles $\boldsymbol{\xi}_t^{(i)}$ are sampled from a forward Markov kernel $\boldsymbol{\xi}_{t+1}^{(i)} \sim K_t(\boldsymbol{\xi}_{t-1}^{(i)}, \boldsymbol{\xi}_t^{(i)})$ to stay close to the target posterior of the next iteration $\tilde{p}_{\beta_{t+1}}(\boldsymbol{\xi}|\mathbf{x}_{1:N}^{\star})$. (iii) The weights of the particles are adjusted based on approximations of (5). As the weight update is typically numerically intractable [14], different SMC-ABC methods [15, 30, 31] have been introduced which propose approximations to the optimal weight update. We refer to Appendix B for a more detailed explanation of SMC-ABC and its different approaches.

## 3 Pseudo-likelihood inference

The proposed PLI methodology, summarized in Figure 1, generalizes the ABC approaches by introducing exponential likelihood kernels with adaptive bandwidth updates, which are motivated from a Variational Inference (VI) perspective.

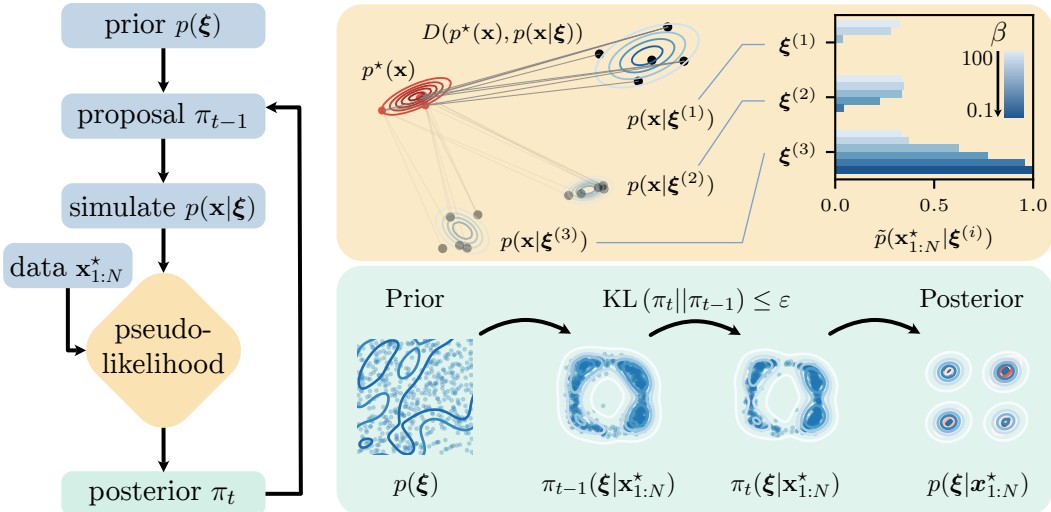

Figure 1: Schematic overview of the introduced iterative Pseudo-Likelihood Inference (PLI) approach. (top) Based on samples drawn from the simulator, the pseudo-likelihood (8) is evaluated based on the discrepancy between the empirical data-generating and likelihood distributions. The bar chart shows how the pseudo-likelihood evaluation changes for different bandwidths. (bottom) The evaluation of the pseudo-likelihood is used to estimate a target posterior $\pi_t$ under trust-region constraints that moves from the prior distribution to the final posterior.

## 3.1 Exponential likelihood kernels

PLI adopts the view of SMC-ABC on approximating a smoothed target posterior $p_t(\boldsymbol{\xi}|\mathbf{x}_{1:N}^\star)$, in the following denoted by $\pi_t(\boldsymbol{\xi})$, by formulating the following constrained VI problem for each inference step $t$,

$$\pi_t(\boldsymbol{\xi}) = \underset{\pi(\boldsymbol{\xi}) \in \mathcal{P}(\boldsymbol{\xi})}{\operatorname{argmin}} \; \mathrm{KL}\left(\pi(\boldsymbol{\xi}) \,||\, p(\boldsymbol{\xi}|\mathbf{x}_{1:N}^\star)\right),$$
$$\text{s.t. } \mathrm{KL}\left(\pi(\boldsymbol{\xi}) \,||\, \pi_{t-1}(\boldsymbol{\xi})\right) \le \varepsilon. \tag{6}$$

The optimization is balanced between fitting the posterior distribution $p(\boldsymbol{\xi}|\mathbf{x}_{1:N}^\star)$ and constraining the information loss between two inference steps $\pi_{t-1}(\boldsymbol{\xi})$ and $\pi_t(\boldsymbol{\xi})$. The loss of information is incorporated as a trust-region constraint with the bound $\varepsilon > 0$ in the space of probability distributions $\mathcal{P}(\boldsymbol{\xi})$ through the Kullback-Leibler (KL) divergence $\mathrm{KL}\left(\pi(\boldsymbol{\xi}) \,||\, \pi_{t-1}(\boldsymbol{\xi})\right)$.

**Theorem 1.** *The optimal target distribution $\pi_t(\boldsymbol{\xi})$ in the optimization problem (6) is given by*

$$\pi_t(\boldsymbol{\xi}) \propto \left(\frac{p(\boldsymbol{\xi})}{\pi_{t-1}(\boldsymbol{\xi})}\right)^{\frac{1}{1+\eta_t}} p(\mathbf{x}_{1:N}^\star|\boldsymbol{\xi})^{\frac{1}{1+\eta_t}} \pi_{t-1}(\boldsymbol{\xi}) \tag{7}$$

*where $\eta_t > 0$ is a dual Lagrangian variable corresponding to the trust-region constraint.*

*Proof.* See Appendix A.1. □

The temperature parameter $\eta_t$ plays the role of an adaptive step size that controls the update step from $\pi_{t-1}(\boldsymbol{\xi})$ to $\pi_t(\boldsymbol{\xi})$. In the limit of small step sizes $\eta_t \to 0$ at convergence, the target posterior (7) turns into the true posterior $\pi_t(\boldsymbol{\xi}) \to p(\boldsymbol{\xi})p(\mathbf{x}_{1:N}^\star|\boldsymbol{\xi})$. In the spirit of ABC-based methods, we approximate the intractable likelihood $p(\mathbf{x}_{1:N}^\star|\boldsymbol{\xi})$ with a Gibbs distribution $\tilde{p}(\mathbf{x}_{1:N}^\star|\boldsymbol{\xi})$, which we call *pseudo-likelihood*, and which is based on a discrepancy measure between the empirical data distribution $p^\star(\mathbf{x}) = 1/N \sum_i \delta_{\mathbf{x}_i^\star}(\mathbf{x})$ and the simulator likelihood $p(\mathbf{x}|\boldsymbol{\xi})$

$$\tilde{p}(\mathbf{x}_{1:N}^\star|\boldsymbol{\xi}) := \frac{1}{Z(\boldsymbol{\xi})} \exp\left(-\frac{D(p^\star(\mathbf{x}), p(\mathbf{x}|\boldsymbol{\xi}))}{2\beta}\right). \tag{8}$$

Here, $Z(\boldsymbol{\xi})$ is the normalization constant, and $\beta > 0$ is a bandwidth parameter that controls the sharpness of the approximation. When the KL divergence is used as the discrepancy measure $D$, we recover the true likelihood, as the following lemma states.

**Lemma 1.** *When $D = \mathrm{KL}$ and $2\beta = 1/N$ in (8), the pseudo-likelihood $\tilde{p}(\mathbf{x}^\star_{1:N}|\boldsymbol{\xi})$ equals the true likelihood $p(\mathbf{x}^\star_{1:N}|\boldsymbol{\xi})$.*

*Proof.* Since $\mathrm{KL}(p^*(\mathbf{x}) \| p(\mathbf{x}|\boldsymbol{\xi})) = -\mathrm{H}[p^*(x)] - N^{-1}\log p(\mathbf{x}^\star_{1:N}|\boldsymbol{\xi})$, then provided $2\beta = 1/N$, the exponential in (8) is given by $\exp(-N\mathrm{KL}(p^*(\mathbf{x}) \| p(\mathbf{x}|\boldsymbol{\xi}))) \propto p(\mathbf{x}^\star_{1:N}|\boldsymbol{\xi})$ and $Z(\boldsymbol{\xi})$ is a constant. $\qquad\square$

However, the KL divergence is intractable in SBI because we cannot evaluate the likelihood. Therefore, we propose replacing the KL divergence with Integral Probability Metrics (IPMs), such as the MMD and the Wasserstein distance, which can be evaluated on distribution samples. Although theoretical analysis is less straightforward in these cases, some results have been obtained in prior works. Consistency and robustness of an MMD-based posterior estimator were shown by Chérief-Abdellatif and Alquier [5] and Wasserstein-based exponential kernels were studied by De Plaen et al. [12]. In this paper, we focus on a practical instantiation of pseudo-likelihood inference, which can accommodate a variety of divergence functions and obtain superior empirical results by leveraging neural posterior approximators and adaptive step-size updates. The following subsections introduce the key components that constitute our method.

## 3.2 Bandwidth adaptation from trust-region principles

The Lagrangian parameter $\eta_t$ has a particularly interesting property. From (7), we see that pulling $\eta_t$ into the pseudo-likelihood (2), yields a *time-dependent tempered pseudo-likelihood*

$$\tilde{p}_t(\mathbf{x}^\star_{1:N}|\boldsymbol{\xi}) := Z(\boldsymbol{\xi})^{1+\eta_t}\exp(-(2\beta_t)^{-1}D(p^\star(\mathbf{x}), p(\mathbf{x}|\boldsymbol{\xi}))) = \tilde{p}(\mathbf{x}^\star_{1:N}|\boldsymbol{\xi})^{\frac{1}{1+\eta_t}}. \tag{9}$$

Here, we introduce the adaptive bandwidth $\beta_t = (1 + \eta_t)\beta$ that approaches $\beta$ in the limit $\eta_t \to 0$. The dual formulation of the stochastic search problem (6) leads to a tractable solution for the optimal Lagrangian parameter $\eta_t$, and hence an optimal bandwidth $\beta_t$ (see Appendix A.1 for more details).

$$g(\eta_t) = -\eta_t\epsilon - (1 + \eta_t)\log \mathbb{E}_{\pi_{t-1}(\boldsymbol{\xi})}\left[\left(\frac{p(\boldsymbol{\xi})}{\pi_{t-1}(\boldsymbol{\xi})}\right)^{\frac{1}{1+\eta_t}}\tilde{p}_t(\mathbf{x}^\star_{1:N}|\boldsymbol{\xi})\right]. \tag{10}$$

While we obtain the primal optimal point in closed form (7) to obtain the optimal dual variable $\eta_t$, we need to resort to numerical optimization of the Lagrangian dual objective. The optimal bandwidth parameter $\beta_t$, that is obtained by maximizing (10), can be seen as an information-bounded trust region update to move the pseudo-likelihood towards the likelihood. In the early inference stages, the proposal prior $p_{t-1}(\boldsymbol{\xi})$ is typically uninformative, and thus the information loss is moderate even if $p_t(\boldsymbol{\xi})$ moves far away from $p_{t-1}(\boldsymbol{\xi})$. In the later inference steps, the proposal distribution is typically pronounced, and small deviations may lead to significant information loss. This intuition suggests that the bandwidth $\beta_t$ should decay over iterations, and indeed Figure 2 shows that $\beta_t$ quickly decays towards zero over a range of values of $\varepsilon$ on the Gaussian location task (Sec.

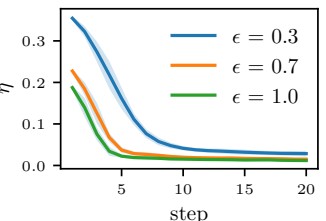

Figure 2: Bandwidth $\eta_t$ recorded for different $\varepsilon$ on the Gaussian location task. The bandwidth is monotonically decreasing over iterations.

C.2). The exact decay schedule of $\beta_t$ is problem-dependent. Therefore, it is convenient to set an information-loss bound $\varepsilon$ and obtain an adaptive bandwidth schedule by optimizing the dual (10) rather than pre-specifying a decay schedule by hand for each problem.

## 3.3 Approximate Bayesian inference with pseudo-likelihoods

Pseudo-Likelihood Inference (PLI) is a sequential SBI methodology based on approximating the target posterior $\pi_t(\boldsymbol{\xi})$ (7). It is closely tied to SMC-ABC by (i) sequentially approximating the target posteriors, (ii) sequentially adapting the bandwidth parameter, and (iii) sequentially updating the proposal distribution for higher sample efficiency. Instead of representing the posterior through a set of weighted particles, the PLI formulation allows for various powerful neural density estimators.

A parameterized density model $q_\phi(\boldsymbol{\xi})$ is trained to approximate the PLI posterior (7) using the m-projection $\min_{\phi\in\Phi}\mathrm{KL}(\pi_t(\boldsymbol{\xi}) \| q_\phi(\boldsymbol{\xi}))$, which results in the Weighted Maximum Likelihood (WML) objective with parameter samples drawn from the proposal prior $\boldsymbol{\xi}^{(k)} \sim \pi_{t-1}(\boldsymbol{\xi}^{(k)})$

$$\max_{\phi\in\Phi} \ \sum_{k=1}^K w^{(k)}\log q_\phi(\boldsymbol{\xi}^{(k)}); \quad w^{(k)} = \left(\frac{p(\boldsymbol{\xi}^{(k)})}{\pi_{t-1}(\boldsymbol{\xi}^{(k)})}\right)^{\frac{1}{1+\eta_t}}\tilde{p}_t(\mathbf{x}^\star_{1:N}|\boldsymbol{\xi}^{(k)}). \tag{11}$$

Thus, we derive a practical PLI algorithm by leveraging this empirical WML objective. Further details on the objective derivation are described in Appendix A.2. Our proposed PLI Algorithm 1 consists of four main steps. First, in lines 4–8, training pairs from the proposal and simulator are drawn, and the discrepancy measure $D(\mathbf{x}_{1:N}^\star, \mathbf{x}_{1:M})$ between the observations and the simulations is evaluated for each $\boldsymbol{\xi}^{(k)}$. We follow Gretton et al. [25] to approximate the MMD between two discrete probability measures, whereas we make use of the entropy regularized optimal transport formulation to approximate the Wasserstein distance [8] (see Appendix C.1). Both versions facilitate parallelization on the GPU. Second, in line 9, the optimal bandwidth under trust region constraint is estimated, and the tempered pseudo-likelihood is evaluated. by maximizing the dual (10). Third, in line 11, the parameterized

**Algorithm 1** Pseudo-Likelihood Inference (PLI)

1: **input:** reference data $\mathbf{x}_{1:N}^\star$, prior $p(\boldsymbol{\xi})$, stochastic simulator $p(\mathbf{x}|\boldsymbol{\xi})$, IPM $D(\cdot, \cdot)$, posterior approximator $q_{\boldsymbol{\phi}}(\boldsymbol{\xi})$, max. iter. $T$
2: initialize proposal prior $\pi_0(\boldsymbol{\xi}) = p(\boldsymbol{\xi})$
3: **for** $t$ in $1\!:\!T$ **do**
4:      sample parameters $\boldsymbol{\xi}^{(1:K)} \sim \pi_{t-1}(\boldsymbol{\xi})$
5:      **for** each $\boldsymbol{\xi}^{(k)}$ **do**
6:          simulate data $\mathbf{x}_{1:M}^{(k)} \sim p(\mathbf{x}|\boldsymbol{\xi}^{(k)})$
7:          compute IPM $s^{(k)} = D(\mathbf{x}_{1:M}^{(k)}, \mathbf{x}_{1:N}^\star)$
8:      **end for**
9:      update $\eta_t$ by maximizing the dual (10)
10:     evaluate $\tilde{p}_t(\mathbf{x}_{1:N}^\star|\boldsymbol{\xi}^{(k)})$ (9)
11:     fit $q_{\boldsymbol{\phi}_t}(\boldsymbol{\xi})$ by WML (11)
12:     set new proposal prior $\pi_t(\boldsymbol{\xi}) = q_{\boldsymbol{\phi}_t}(\boldsymbol{\xi})$
13: **end for**
14: **output:** approximate posterior $q_{\boldsymbol{\phi}_T}(\boldsymbol{\xi})$

density estimator $q_{\boldsymbol{\phi}}(\boldsymbol{\xi})$ is trained to approximate the target posterior $\pi_t(\boldsymbol{\xi})$ via the m-projection (11). Note that the expectation w.r.t. the proposal distribution $\pi_{t-1}(\boldsymbol{\xi})$ enables gradient descent on the $q_{\boldsymbol{\phi}_t}$ estimator without requiring a differentiable simulator. Fourth, in line 12, we set the current posterior approximation $q_{\boldsymbol{\phi}_t}(\boldsymbol{\xi})$ as the proposal $\pi_t(\boldsymbol{\xi})$ for the next inference step, thus leveraging bootstrapping of the density estimator. While we restrict the analysis in this paper to the m-projection, we note that the i-projection can also be employed, as shown in Appendix A.3.

**Normalization.** The normalization term $Z(\boldsymbol{\xi})$ in the definition of the pseudo-likelihood (8) requires taking an integral over the reference data $\mathbf{x}_{1:N}^\star$, which is infeasible in practice. When the KL divergence is used in the kernel, $Z(\boldsymbol{\xi})$ does not depend on $\boldsymbol{\xi}$, as shown in Lemma 1. While in general, the dependence on $\boldsymbol{\xi}$ cannot be neglected, its influence on the weights in (10) and (11) may be negligible, provided the relative ranking of the samples is not affected significantly. In Appendix A.4, we provide an ablation study on low-dimensional problems where the integral over $\mathbf{x}_{1:N}^\star$ can be approximated by sampling. We observed that, even though the ranking correlation of the weights $w^{(k)}$ in (11) is different with and without estimating $Z(\boldsymbol{\xi})$, the final posterior is not affected. Therefore, in the subsequent experiments, we treat it as a constant, as in the ideal case of the KL divergence. Nevertheless, this is a point where our practical implementation does not follow the theoretical derivation strictly, and this issue should be addressed in future work.

## 4 Experiments

We compare the PLI framework against SMC-ABC [15] and APT [24] on five diverse tasks. Our implementation is based on Wasserstein-ABC [3], but instead of the employed r-hit kernel [30], our implementation is based on population Monte Carlo (Alogrithm 3 [31]) because we observed improved performance in preliminary studies. A summary of the different ABC methods is given in Appendix B and Table A.1. APT was chosen as the representative for the class of SNPE algorithms. We leverage Neural Spline Flows (NSFs) [19] as density estimators for both PLI and APT. Both neural flow configurations share the same base network architecture, but for APT, the conditional flow is augmented with an embedding network (Appendix C). All experiments are implemented in JAX [4], and each ran on a single Nvidia RTX 3090.[1] To make the experiments comparable, the simulation budgets of PLI and APT were fixed to 5000 samples per inference step over 20 episodes, while ABC ran for 200 episodes on 1000 particles.

### 4.1 Evaluation metrics

The model is compared against the reference posterior samples $\boldsymbol{\xi}^\star$ when available. We also quantify the methods' performances based on their realizations by computing the Wasserstein distance and

---

[1]https://github.com/theogruner/pseudo_likelihood_inference

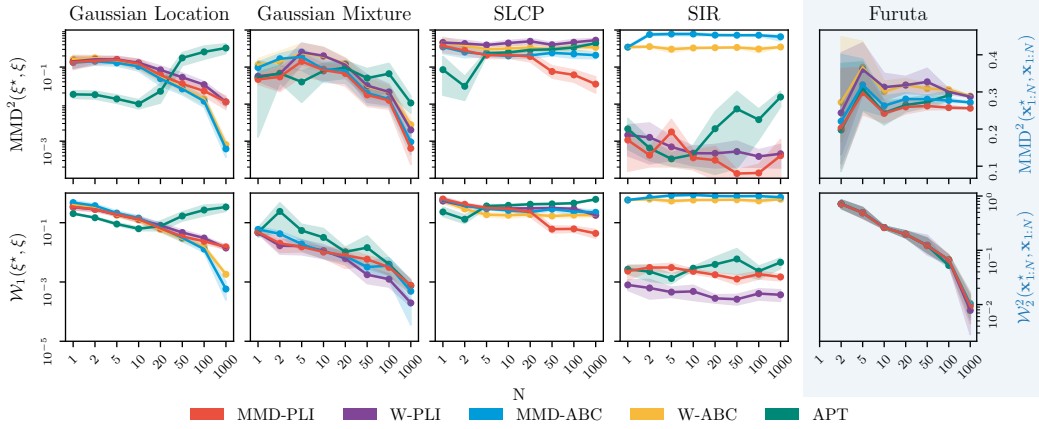

Figure 3: Evaluation of the posterior performance on five different tasks. We report the mean and 95% ci over 10 random seeds, each carried out using $N$ data points for conditioning. We compare samples from the approximate posterior $\boldsymbol{\xi} \sim q(\boldsymbol{\xi}|\mathbf{x}_{1:N}^{\star})$ against reference posterior samples using MMD and the Wasserstein distance. No posterior samples were available for the Furuta pendulum. Therefore, the performance is evaluated in the observation space. Lower values are better for all metrics. PLI is the preferred method for conditioning on multiple observations due to its steady improvement with increasing $N$. ABC performs better than PLI on Gaussian Location and Gaussian Mixture tasks but lags in more complex tasks. APT excels with few observations but degrades as $N$ increases.

the MMD. The comparison is carried out on 10 000 samples each. Furthermore, we use Posterior Predictive Checks (PPCs) to evaluate the predictive capabilities of the posterior models in the observation space $\mathbb{E}_{q(\boldsymbol{\xi}|\mathbf{x}_{1:N}^{\star})}[D(\mathbf{x}_{1:N}^{\star}, \mathbf{x}_{1:M})]$. Due to the computational limits, the PPCs are carried out on 1000 simulations against the reference data. Lueckmann et al. [32] also report results with classifier-based tests and the kernelized Stein-discrepancy. However, since benchmarking is not our focus, we restrict the analysis to comparing with the Wasserstein and MMD.

## 4.2 Tasks

We evaluate PLI on four common benchmarking tasks within the SBI community [32]: Gaussian Location, a Gaussian Mixture Model, Simple-Likelihood Complex-Posterior, and SIR. Further, we add a system identification task on a Furuta pendulum representing a highly dynamic continuous control system. The tasks' specifications are listed in Appendix C. For each task, we conduct experiments for different numbers of available reference observations $N = \{1, 2, 5, 10, 20, 50, 100, 1000\}$. The reference observations are simulated based on a pre-defined ground-truth parameter $\boldsymbol{\xi}^{\mathrm{gt}}$. Although PLI and ABC can cope with varying numbers of observations $N$ and numbers of simulations per parameter $M$, we choose $N = M$ for all experiments since it is required by APT. In the following paragraphs, we first discuss the results of the benchmarking tasks and present a separate discussion for the Furuta pendulum. Figure 3 gives a quantitative overview of the benchmarking tasks compared to the reference posteriors, while Figure C.1 in the Appendix complements the study by showing the posterior predictive performances.

**Benchmarking tasks.** Each benchmarking task presents different challenges that must be addressed by the SBI methods. In Gaussian Location, the task is to infer a uni-modal 10-dimensional Gaussian distribution. Gaussian Mixture Model and SLCP feature multi-modal posteriors that require flexible density estimators. SIR is a well-known epidemiological model that features 10-dimensional data of the dynamical system. All methods generally depict a reoccurring behavior on the different benchmarking tasks, as shown in Figure 3. For fewer observations ($N \lesssim 20$), APT matches the reference posterior better than the other approaches, whereas ABC and PLI match the posterior data better with increasing $N$. In particular, PLI consistently improves with an increasing number of reference samples. The influence of $N$ on the shape of the posterior is further visualized in Figure C.2, which compares the posterior approximations of all methods for $N = 2$ and $N = 100$ reference observations. For the SLCP task, ABC struggles to capture the multi-modality of the SLCP task. This effect is further illustrated when comparing Figures C.3 and C.4, which allow for a qualitative

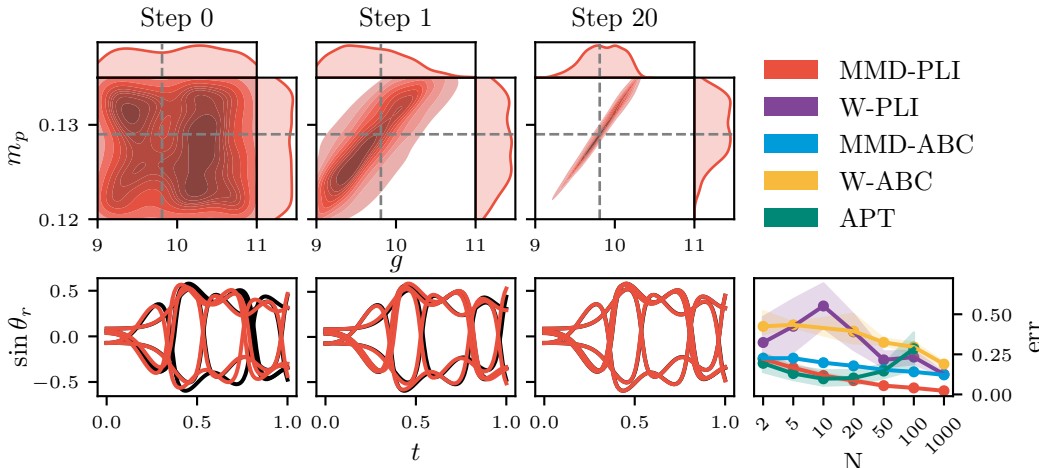

Figure 4: Empirical and quantitative evaluation on the Furuta pendulum. (top) Snapshots of the posterior evolution on the $g - m_p$ plane on the Furuta pendulum for $N = 1000$. (bottom) Predictive performance of the learned MMD-PLI posterior for the angular rotation $\sin \theta_r$. The stochasticity of the simulator is removed by synchronizing the initial state between the reference and predicted simulations. Thus, the only discrepancies between trajectories are due to the model not capturing the dynamics parameters of the system. After the inference has been completed (step 20), the predictive simulator (— MMD-PLI) can completely recover the ground truth dynamics (— Reference). (right) Evaluation of the mean accumulated error over 1000 trajectories with synchronized initial states between the simulation and the reference trajectory. All approaches improve with rising $N$ while PLI with MMD matches the reference data best.

assessment of the posterior approximations. On SIR, PLI variants show significantly improved performance compared to the other baselines. Generally, we find that ABC and PLI perform better with MMD than with Wasserstein distance. This observation can be attributed to the Wasserstein distance not scaling well to high dimensions, which has been reported by recent studies [17, 16].

**Furuta pendulum.** The Furuta pendulum is an inverted double pendulum setup [21]. While the system's dynamics are inherently deterministic, small perturbations of the initial state around its unstable equilibrium point lead to highly diverse trajectories. The observation space is $T \times 6$ dimensional, where $T$ is the number of time steps per trajectory. We set the sampling frequency of the simulation to $100\,\mathrm{Hz}$ and the duration to $1\,\mathrm{sec}$, resulting in 600-dimensional observations. No reference posterior is available for this task; thus, the analysis is restricted to quantifying the observed data. Given the similarity of the Wasserstein distance and the MMD in parameter and observation space on the previous tasks, we argue that a comparison based on PPCs, i.e., $\mathcal{W}_2^2(\mathbf{x}_{1:N}^\star, \mathbf{x}_{1:M})$ and $\mathrm{MMD}^2(\mathbf{x}_{1:N}^\star, \mathbf{x}_{1:M})$, is sufficient. In Figure 3, the Wasserstein PPC and MMD PPC exhibit divergent behaviors, with the former indicating enhanced performance as reference observations increase, in contrast to the moderate improvement suggested by MMD. Therefore, we evaluate the models' predictive performances on the deterministic system by synchronizing the initial states of the reference data and the simulations in Figure 4. This modification ensures that the similarity between two rollouts can be evaluated by the accumulated error $\mathrm{err} = \sum_i |x_i^\star - x_i|$. While all approaches perform better with more reference observations, MMD-PLI matches the reference dynamics best. Additionally, MMD is favored here, as MMD-based PLI and ABC outperform their Wasserstein counterparts, with both the MMD PPC plot and error plot showcasing congruent trends for $N \geq 20$. The appended posterior plots in Figures C.5 and C.6 reveal that for $N = 2$, all methods are widely spread over the prior region, yet converge to the ground truth. However, APT cannot recover the ground truth for $N \geq 100$, whereas PLI and ABC center around the ground truth.

## 5   Related work

In the previous sections, we have seen that PLI is algorithmically similar to ABC methods with SMC samplers. Therefore, approximating the likelihood by the empirical pseudo-likelihood (2) enables

drawing from the rich toolbox of existing approximate inference algorithms. This section introduces related research fields and shows how PLI fits among them.

**Sequential neural density estimation.** With the enriched class of neural density estimators, amortized SBI methods have received increasing interest in recent years. Similar to ABC, synthetic samples from the simulator are used to approximate the posterior. Sequential neural density estimation methods can be further classified into methods that directly train a posterior estimator [40, 24], a neural likelihood [41, 22], or a neural ratio estimator [19, 36]. All methods have in common that they do not rely on an approximation of the posterior model but are optimized solely on pairs of parameter samples from a proposal distribution $\boldsymbol{\xi}^{(k)} \sim p_t(\boldsymbol{\xi})$ and its corresponding simulation $\mathbf{x}^{(k)} \sim p(\mathbf{x}|\boldsymbol{\xi}^{(k)})$. We note that the original papers have only reported posteriors conditioned on a single observation $\mathbf{x}$. While technically, these methods can incorporate multiple data points, this requires either stacking multiple observations or falling back to summary statistics. As noted in [50], neural likelihood estimators can sidestep these requirements by evaluating the log-likelihood of single observations and carrying out MCMC sampling on the joint log-likelihood. Yet, leveraging the neural likelihood restricts the evaluation of the posterior.

**Summary statistics.** ABC has commonly relied on reducing the dimensionality of the raw observations with summary statistics [48]. These summaries must be carefully chosen and are often task-specific, restricting the general applicability of ABC. Recent additions to ABC methods report on replacing summary statistics with statistical distances [17]. While direct comparison of the raw data suffers from the curse of dimensionality, comparing the observations through empirical measures sidesteps this issue [17]. Bernton et al. [3] report on augmenting the likelihood kernel with the Wasserstein distance, while Park et al. [42] leverage the kernelized approximation of the MMD [25]. Other contributions include the Cramér-von-Mises distance [20] and the energy distance [39]. While statistical distances are appealing due to their general applicability, Drovandi and Frazier [17] conclude that they are limited by their high computational requirements. Approaches proposed for automated summary design include ABC with indirect inference, which utilizes an auxiliary model to evaluate data summaries [23, 18].

**Particle mirror descent.** A posterior updating similar to ours (7) has been derived in Particle Mirror Descent (PMD) [10]. PMD tackles particle depletion by incorporating the proposal distribution of the previous round into the optimization process. Furthermore, the authors show that the proposed method converges to the posterior given $m$ posterior samples by $\mathcal{O}(1/\sqrt{m})$. Our version can be seen as extending their approach to the case of intractable likelihoods. We extend PMD to neural density estimators using samples from the proposal posterior (7) as a training set.

**Geometric path and likelihood tempering.** Rewriting the optimal posterior (7) reveals a close relation of the optimal PLI posterior (4) and the geometric path formulation [6, p. 335], $\pi_t(\boldsymbol{\xi}) \propto \pi_{t-1}^{1-\lambda}(\boldsymbol{\xi}) \, p(\mathbf{x}_{1:N}^\star, \boldsymbol{\xi})^\lambda$. The optimal posterior moves from the proposal distribution $\pi_t(\boldsymbol{\xi})$ at inference time $t$ to the target posterior $\pi_t(\boldsymbol{\xi}|\mathbf{x}_{1:N}^\star) \propto p(\mathbf{x}_{1:N}^\star, \boldsymbol{\xi})$ along the geometric path that is parameterized by $\lambda$. The formulation differentiates from likelihood tempering in SMC samplers [6] by leveraging the proposal instead of the prior distribution. Note, however, that for $t = 0$, the proposal mimics the prior, and thus the PLI geometric path has the same boundary values as in classical likelihood tempering. While the tempered posterior cannot be applied to SMC samplers due to its dependence on the proposal distribution, the geometric path formulation based on the prior distribution gives rise to sequential annealing ABC [1].

**Generalized variational inference.** Introduced by Knoblauch et al. [29], Generalized Variational Inference (GVI) is an extension of the standard variational inference framework that starts from the optimization view of Bayes' rule and generalizes it by considering different losses, divergences, and variational families. Therefore, various Bayesian inference methods can be seen as instantiations of GVI with different choices of these three parameters. In particular, ABC and PLI can be seen as GVI with the choices $P(K_\beta(\mathbf{x}_{1:N}^\star, \mathbf{x}_{1:M}), \mathrm{KL}, \mathcal{P}(\Theta))$ since they employ a likelihood kernel as a loss. Crucially, specifying a different loss function instead of the typical log-likelihood can be shown to add robustness against model misspecification [29]. PAC-Bayes [26] can be seen as a generalization of ABC as it covers the whole space of possible loss functions $l(\mathbf{x}_{1:N}^\star, \boldsymbol{\xi})$. The PAC-Bayesian theory provides a broad array of risk bounds for generalized Bayesian learning methods.

## 6  Conclusion

We propose Pseudo-Likelihood Inference (PLI), a new addition to the toolbox of SBI methods. PLI is targeted for Bayesian inference tasks in which the posterior is conditioned on multiple observations simultaneously. For that, we derive a softened ABC posterior from a constrained variational inference problem and leverage IPMs between the empirical observations to assess the intractable likelihood. The derived posterior formulation enables the learning of flexible neural density estimators from black-box simulators, extending the range of applicability for ABC methods. Our experiments assess how well PLI, ABC, and SNPE perform based on their generative power when given varying amounts of reference observations as a condition. Given few observations, SNPE-based methods perform better than ABC and PLI, which rely on statistical distances. However, when more data is available, ABC and PLI methods perform better. When the posterior distribution is simple, ABC is efficient at reproducing it using fast particle updates. However, PLI is a better option for complex posterior distributions because of its more adaptable neural density estimator. Additionally, PLI evaluates the posterior probability, which is useful in downstream tasks that require uncertainty quantification.

**Limitations.**  In PLI, the computational cost is distributed among three main computations: the simulation, the summary statistics estimation, and the normalizing flow training. While the computational effort is large for every computation, PLI leverages parallelization on GPUs for all three computations. In the provided experiments, the training process of the neural model takes the main computational budget, while simulation and summary statistics are negligible. On a Nvidia RTX 3090, ABC typically runs 2-10 min, while PLI and SNPE take 60-90 min, depending on the task. Simpler models however, such as multivariate Gaussian or GMM, reduce the computation time to the simulation. Furthermore, we would like to express that high-fidelity simulators might increase the simulation time significantly, making the simulation the most costly operation within the inference pipeline. Instead of utilizing IPMs, one could exploit adversarial strategies to approximate the KL divergence, as explored by Mescheder et al. [35] and Santana and Hernández-Lobato [45].

## Acknowledgements

This work was funded by the Hessian Ministry of Science and the Arts (HMWK) through the projects "The Third Wave of Artificial Intelligence - 3AI", hessian.AI, and the grant "Einrichtung eines Labors des Deutschen Forschungszentrum für Künstliche Intelligenz (DFKI) an der Technischen Universität Darmstadt". Fabio Muratore was employed by the Robert Bosch GmbH during the time of this collaboration.

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

# A  Algorithmic details: pseudo-likelihood inference

## A.1  Deriving the optimal PLI parameter distribution

**Theorem 1.** *The optimal target distribution $\pi_t(\boldsymbol{\xi})$ in the optimization problem* (6) *is given by*

$$\pi_t(\boldsymbol{\xi}) \propto \left( \frac{p(\boldsymbol{\xi})}{\pi_{t-1}(\boldsymbol{\xi})} \right)^{\frac{1}{1+\eta_t}} p(\mathbf{x}_{1:N}^\star|\boldsymbol{\xi})^{\frac{1}{1+\eta_t}} \pi_{t-1}(\boldsymbol{\xi}) \tag{12}$$

*where $\eta_t > 0$ is a dual Lagrangian variable corresponding to the trust-region constraint.*

*Proof.* The solution to the stochastic search problem (6) can be obtained from Lagrangian optimization. The optimization problem is restated here for readability

$$\pi_t(\boldsymbol{\xi}) = \arg\min_{\pi(\boldsymbol{\xi})} \mathrm{KL}\left(\pi(\boldsymbol{\xi}) \,||\, p(\boldsymbol{\xi}|\mathbf{x}_{1:N}^\star)\right),$$

$$\text{s.t. } \mathrm{KL}\left(\pi(\boldsymbol{\xi}) \,||\, \pi_{t-1}(\boldsymbol{\xi})\right) \leq \varepsilon,$$

$$\int \pi(\boldsymbol{\xi})\, \mathrm{d}\boldsymbol{\xi} = 1.$$

We decompose the KL objective into two terms by applying Bayes' rule

$$\mathrm{KL}\left(\pi(\boldsymbol{\xi}) \,||\, p(\boldsymbol{\xi}|\mathbf{x}_{1:N}^\star)\right) = - \mathop{\mathbb{E}}_{\pi(\boldsymbol{\xi})}\left[\log p(\mathbf{x}_{1:N}^\star|\boldsymbol{\xi})\right] + \mathrm{KL}\left(\pi(\boldsymbol{\xi}) \,||\, p(\boldsymbol{\xi})\right). \tag{13}$$

The constrained optimization problem (6) can be reformulated with Lagrange multipliers as

$$
\begin{aligned}
\mathcal{L}(\pi) = {}& - \int \pi(\boldsymbol{\xi}) \log p(\mathbf{x}_{1:N}^\star|\boldsymbol{\xi}) \mathrm{d}\boldsymbol{\xi} \\
& + \int \pi(\boldsymbol{\xi}) \log \frac{\pi(\boldsymbol{\xi})}{p(\boldsymbol{\xi})} \mathrm{d}\boldsymbol{\xi} \\
& + \eta \left( \int \pi(\boldsymbol{\xi}) \log \frac{\pi(\boldsymbol{\xi})}{\pi_{t-1}(\boldsymbol{\xi})} \mathrm{d}\boldsymbol{\xi} - \varepsilon \right) \\
& + \lambda \left( \int \pi(\boldsymbol{\xi}) \mathrm{d}\boldsymbol{\xi} - 1 \right) \\
= {}& \int \pi(\boldsymbol{\xi}) \left[ -\log p(\mathbf{x}_{1:N}^\star|\boldsymbol{\xi}) + \log \frac{\pi(\boldsymbol{\xi})}{p(\boldsymbol{\xi})} + \eta \log \frac{\pi(\boldsymbol{\xi})}{\pi_{t-1}(\boldsymbol{\xi})} + \lambda \right] \mathrm{d}\boldsymbol{\xi} - \eta\varepsilon - \lambda. \tag{14}
\end{aligned}
$$

Here, we leveraged the assumption that the likelihood $p(\mathbf{x}_{1:M}|\boldsymbol{\xi})$ is fixed for all joint distributions, and thus, the joint distributions can be split into the likelihood $p(\mathbf{x}_{1:N}^\star|\boldsymbol{\xi})$ and their associated prior/proposal distributions. The gradient of the Lagrangian vanishes for the optimal parameter distribution

$$\frac{\partial \mathcal{L}}{\partial \pi}\bigg|_{\pi=\pi_t} = -\log p(\mathbf{x}_{1:N}^\star|\boldsymbol{\xi}) + \left[\log \frac{\pi_t(\boldsymbol{\xi})}{p(\boldsymbol{\xi})} + 1\right] + \eta \left[\log \frac{\pi_t(\boldsymbol{\xi})}{\pi_{t-1}(\boldsymbol{\xi})} + 1\right] + \lambda = 0. \tag{15}$$

Reformulation yields

$$
\begin{aligned}
\pi_t(\boldsymbol{\xi}) &= p^{\frac{1}{1+\eta}}(\boldsymbol{\xi})\, \pi_{t-1}^{\frac{\eta}{1+\eta}}(\boldsymbol{\xi})\, \exp\left( \frac{\log p(\mathbf{x}_{1:N}^\star|\boldsymbol{\xi})}{1+\eta} - \frac{1+\eta+\lambda}{1+\eta} \right) \\
&= Q^{-1}(\eta) \left( \frac{p(\boldsymbol{\xi})}{\pi_{t-1}(\boldsymbol{\xi})} \right)^{\frac{1}{1+\eta}} \exp\left( \frac{\log p(\mathbf{x}_{1:N}^\star|\boldsymbol{\xi})}{1+\eta} \right) \pi_{t-1}(\boldsymbol{\xi}).
\end{aligned} \tag{16}
$$

The normalization constant $Q(\eta) = \exp((1+\eta+\lambda)/(1+\eta))$ follows by marginalization of (16)

$$Q(\eta) = \mathop{\mathbb{E}}_{\pi_{t-1}(\boldsymbol{\xi})}\left[ \left( \frac{p(\boldsymbol{\xi})}{\pi_{t-1}(\boldsymbol{\xi})} \right)^{\frac{1}{1+\eta}} \exp\left( \frac{\log p(\mathbf{x}_{1:N}^\star|\boldsymbol{\xi})}{1+\eta} \right) \right].$$

We further obtain the dual of the Lagrangian by reinserting (16) into the Lagrangian (14)

$$g(\eta) = -\eta\varepsilon - (1+\eta)\log(Q(\eta)). \tag{17}$$

$$\square$$

## A.2 Weighted maximum likelihood optimization (m-projection)

The m-projection of the optimal posterior onto the approximation family results in a weighted maximum likelihood formulation

$$\min_{\phi} \ \mathrm{KL}\left(\pi_t(\boldsymbol{\xi}) \ || \ q_{\phi}(\boldsymbol{\xi})\right) \tag{18}$$

$$= \max_{\phi} \int \log q_{\phi}(\boldsymbol{\xi}) \, \pi_t(\boldsymbol{\xi}) \, \mathrm{d}\boldsymbol{\xi}$$

$$= \max_{\phi} \int \frac{1}{Q} \left(\frac{p(\boldsymbol{\xi})}{\pi_{t-1}(\boldsymbol{\xi})}\right)^{\frac{1}{1+\eta_t}} \tilde{p}(\mathbf{x}_{1:N}^{\star}|\boldsymbol{\xi})^{\frac{1}{1+\eta_t}} \, \pi_{t-1}(\boldsymbol{\xi}) \log q_{\phi}(\boldsymbol{\xi}) \, \mathrm{d}\boldsymbol{\xi}$$

$$= \max_{\phi} \ \mathbb{E}_{\pi_{t-1}(\boldsymbol{\xi})} \left[ \underbrace{\left(\frac{p(\boldsymbol{\xi})}{p_{t-1}(\boldsymbol{\xi})}\right)^{\frac{1}{1+\eta_t}} \tilde{p}(\mathbf{x}_{1:N}^{\star}|\boldsymbol{\xi})^{\frac{1}{1+\eta_t}}}_{w} \log q_{\phi}(\boldsymbol{\xi}) \right]. \tag{19}$$

The weighting term $w$ is independent of $\phi$, and as such, the formulation facilitates optimizing neural density estimators with gradient descent. This optimization resolves to weighted maximum likelihood where the weights are obtained from the pseudo-likelihood (8) and the importance weights. The weighted maximum likelihood formulation can be optimized in closed form for linear Gaussian models [43, 13], with Expectation-Maximization (EM) using Gaussian Mixture Models (GMMs) [11] or with gradient descent as done in this paper.

## A.3 Optimizing with the i-projection

We can reformulate the minimization problem for the i-projection in the following way

$$\min_{\phi} \ \mathrm{KL}\left(q_{\phi}(\boldsymbol{\xi}) \ || \ \pi_t(\boldsymbol{\xi})\right) \tag{20}$$

$$= \min_{\phi} \ \mathbb{E}_{q_{\phi}(\boldsymbol{\xi})} \left[ \log \frac{q_{\phi}(\boldsymbol{\xi})}{Q_{\phi}^{-1} \left(\frac{p(\boldsymbol{\xi})}{p_{t-1}(\boldsymbol{\xi})}\right)^{1/(1+\eta_t)} \tilde{p}(\mathbf{x}_{1:N}^{\star}|\boldsymbol{\xi})^{\frac{1}{1+\eta_t}} \, \pi_{t-1}(\boldsymbol{\xi})} \right] \tag{21}$$

$$= \min_{\phi} \ \mathrm{KL}\left(q_{\phi}(\boldsymbol{\xi}) \ || \ \pi_{t-1}(\boldsymbol{\xi})\right) - \mathbb{E}_{\pi_{t-1}(\boldsymbol{\xi})} \left[ \frac{q_{\phi}(\boldsymbol{\xi})}{\pi_{t-1}(\boldsymbol{\xi})} \log \left( \frac{\left(\frac{p(\boldsymbol{\xi})}{p_{t-1}(\boldsymbol{\xi})}\right)^{1/(1+\eta)} \tilde{p}(\mathbf{x}_{1:N}^{\star}|\boldsymbol{\xi})^{\frac{1}{1+\eta_t}}}{Q_{\phi}} \right) \right].$$

The equation above alleviates the issue of back-propagating through the simulator by using importance sampling. The optimization problem is fitting the posterior estimator $q$ to the proposal $p(\boldsymbol{\xi})$ while having a regularization term that forces the distribution to fit the reference data. The temperature parameter $\eta$ can thus be interpreted as weighting the regularization term. Small values of $\eta$ put more emphasis on the regularization term, while large values concentrate on containing the information of the proposal. For linear Gaussian models, a closed-form expression for the KL term in (21) exists. Therefore, it can be directly optimized using any non-linear optimization method.

Table A.1: Kernels used for the ABC and PLI variants during the experiments in Section 4. An IPM denoted by $D$ plays the role of a distance measure between the reference data $\mathbf{x}_{1:N}^{\star}$ and simulated samples $\mathbf{x}_{1:M}$. Parameter $\beta_t$ controls the kernel bandwidth (see Section 3.1). Effective Sample Size (ESS) is defined as the inverse of the normalized weight's variance.

| $K_{\beta_t}(D(\mathbf{x}_{1:M}, \mathbf{x}_{1:N}^{\star}))$ | Algorithm | $\beta_t$ estimation | Update |
|---|---|---|---|
| $\mathbb{1}_{\{D(\mathbf{x}_{1:M}, \mathbf{x}_{1:N}^{\star}) \leq \beta_t\}}$ | SMC ABC | ESS | $\beta_t^{\star} = \arg\min_{\beta_t} \ \mathrm{ESS}(w_t, \beta_t) - \alpha\mathrm{ESS}(w_{t-1}, \beta_{t-1})$ |
| | PMC ABC | $\alpha$-Quantile | $\beta_t^{\star} = Q_{D(\mathbf{x}_{1:M}^t, \mathbf{x}_{1:N}^{\star})}(\alpha)$ |
| $\exp\left(-\frac{D(\mathbf{x}_{1:M}, \mathbf{x}_{1:N}^{\star})}{2\beta_t}\right)$ | PLI | Trust-region | $\beta_t^{\star} = \beta(1 + \arg\max_{\eta_t} g(\eta_t))$, see (10) |

## A.4 Analysis of the partition function

We shed some light on the intractable log-partition function $Z(\boldsymbol{\xi})$ introduced in the pseudo-likelihood (8). The partition function $Z(\boldsymbol{\xi})$ of (8) is an integral over sample space of $\mathbf{x} \in \mathcal{X}$

$$Z(\boldsymbol{\xi}) = \int_{\mathcal{X}} \exp\left(-\frac{D(\mathbf{x}^{\star}_{1:N}, \mathbf{x}_{1:M})}{\beta}\right) \, \mathrm{d}\mathbf{x}^{\star}_{1:N}. \tag{22}$$

To approximate the intractable quantity, we approximate the integral through Monte Carlo simulations with a uniform distribution $\mathcal{U}$

$$Z(\boldsymbol{\xi}) \approx \frac{V}{N} \sum_{i=1}^{N} \exp\left(-\frac{D((\mathbf{x}^{\star}_{1:N})_i, \mathbf{x}_{1:M})}{\beta}\right); \quad (\mathbf{x}^{\star}_{1:N})_i \sim \mathcal{U}(\cdot; \bar{\mathbf{x}} - 5\sqrt{\beta}, \bar{\mathbf{x}} + 5\sqrt{\beta}). \tag{23}$$

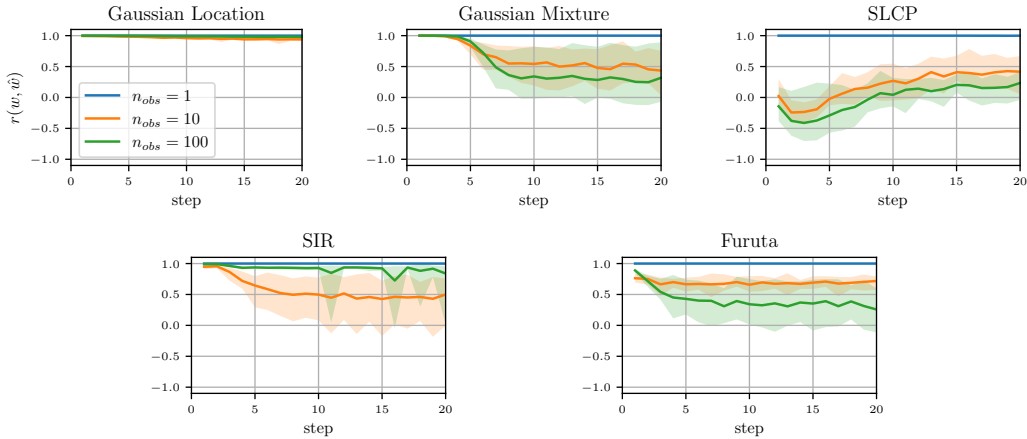

Figure A.1: Spearman correlation coefficient $r(w, \hat{w}) \in [-1, 1]$ between the partition corrected weights $\hat{w}_i$ and the uncorrected weights $w_i$. High values of $r$ correspond to a high correlation of the weight rankings, thus meaning that the weights preserve relative ordering.

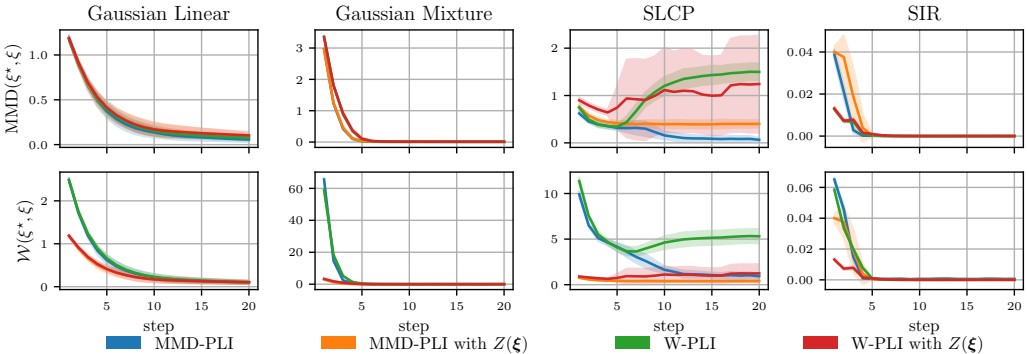

Figure A.2: Training plots comparing PLI trained with and without the partition function $Z(\boldsymbol{\xi})$ on $N = 100$ samples. Top row: MMD between posterior samples and model samples. Bottom row: Wasserstein distance between posterior samples and model samples. On all tasks but SLCP, the inclusion of the partition function does not change the posterior inference.

As we cannot sample over the whole space $\mathcal{X}$, we choose to sample over the $5\sigma$ interval of the exponential kernel, where $\bar{\mathbf{x}}$ represents the mean of 100 prior simulations. We use 10000 samples from $\mathcal{U}$ to approximate the partition function and evaluate the $Z(\boldsymbol{\xi})$ based on 100 samples from the target posterior $\pi_t(\boldsymbol{\xi})$.

We evaluate the influence of the partition function $Z(\boldsymbol{\xi})$ on the performance of our PLI algorithm by comparing the weights $\hat{w}_i$ and $w_i$ as defined in (11) with and without $Z(\boldsymbol{\xi})$, respectively. Despite the

weights being numerically different, we hypothesize that they do not change the relative ordering of the samples and, therefore, do not affect the Weighted Maximum Likelihood (WML) update (11) significantly. Therefore, we employ the Spearman correlation coefficient, a nonparametric measure of rank correlation, to capture such dependencies, which we report in Figure A.1. Contrary to our hypothesis, the weights with and without normalization are not perfectly rank-correlated in four out of five experiments when there is more than one observed data point. Potentially, this could lead to a different WML update, but when we evaluate the whole PLI algorithm with and without the partition function $Z(\boldsymbol{\xi})$, we find that its influence is marginal for both MMD-PLI and W-PLI in Figure A.2. Only on the SLCP task, an improvement of W-PLI can be seen when using the partition function, whereas MMD-PLI performs even better without it.

We have motivated PLI from a VI perspective. As mentioned in the related work section (see Sec. 5), this is similar to the General Variational Inference (GVI) approach [29], which also defines the posterior as a solution to an optimization problem, namely

$$\pi(\boldsymbol{\xi}) = \underset{\pi}{\arg\min} \, \beta N \underset{\pi(\boldsymbol{\xi})}{\mathbb{E}} \left[ \mathcal{L}_N(\boldsymbol{\xi}) \right] + \mathrm{KL} \left( \pi(\boldsymbol{\xi}) \, || \, p(\boldsymbol{\xi}) \right). \tag{24}$$

When the loss is defined as the log-likelihood, $\mathcal{L}_N(\boldsymbol{\xi}) = \log p(\mathbf{x}_{1:N}^\star | \boldsymbol{\xi})$ with $\beta = 1/N$, the PLI objective (6) is recovered. However, when we substitute the pseudo-likelihood (8) into the GVI objective, we obtain an additional term in the loss which is given as an expectation over the log-partition function $\mathbb{E}_\pi[-\log Z(\boldsymbol{\xi})]$. Therefore, PLI can be seen as GVI with an additional loss term, or GVI can be equated with PLI using unnormalized pseudo-likelihood.

## B    Algorithmic details: sequential Monte Carlo ABC

The foundations of SMC-ABC have been laid by Del Moral et al. [14], who introduced SMC samplers. These samplers describe an approximate inference routine in which the posterior is approximated through a sequence of intermediate target posteriors. In the context of ABC, the sequence of intermediate posteriors is defined by an adaptive bandwidth $\beta_t$ of the approximate posterior $p_{\beta_t}(\boldsymbol{\xi} | \mathbf{x}_{1:N}^\star)$ (4). Additionally, the sample efficiency of ABC is improved by replacing the prior as the sampling distribution with a proposal distribution $\pi_t(\boldsymbol{\xi})$. The proposal distribution is represented by a set of particles $\pi_t(\boldsymbol{\xi}) = 1/M \sum_{i=1}^M \delta_{\boldsymbol{\xi}_t^{(i)}}(\boldsymbol{\xi})$ and through importance sampling an approximation of the target posterior $p_{\beta_t}(\boldsymbol{\xi} | \mathbf{x}_{1:N}^\star)$ can be obtained, (see (5)),

$$p_{\beta_t}(\boldsymbol{\xi} | \mathbf{x}_{1:N}^\star) \approx q_t(\boldsymbol{\xi}) = \sum_{i=1}^M W_t^{(i)} \delta_{\boldsymbol{\xi}_t^{(i)}}(\boldsymbol{\xi}); \quad W_t^{(i)} = \frac{p_{\beta_t}(\boldsymbol{\xi}_t^{(i)} | \mathbf{x}_{1:N}^\star)}{\pi_t(\boldsymbol{\xi}^{(i)})}, \tag{25}$$

where $W_t^{(i)}$ denote the importance weights. The proposal distribution $\pi_t(\boldsymbol{\xi})$ should ideally stay close to the target posterior $p_{\beta_t}(\boldsymbol{\xi} | \mathbf{x}_{1:N}^\star)$ to improve the sample efficiency. Therefore, the proposal distribution is updated based on a Markov kernel $K_{t+1}(\boldsymbol{\xi}_t, \boldsymbol{\xi}_{t+1})$ which is the transition probability from $\boldsymbol{\xi}_t$ to $\boldsymbol{\xi}_{t+1}$. The update of the proposal distribution is typically numerically intractable as it requires marginalization, i.e., integration over $\boldsymbol{\xi}_t$ for each inference step $0 : t$

$$\pi_{t+1}(\boldsymbol{\xi}_{t+1}) = \int K_{t+1}(\boldsymbol{\xi}_t, \boldsymbol{\xi}_{t+1}) \pi_t(\boldsymbol{\xi}_t) \mathrm{d}\boldsymbol{\xi}_t. \tag{26}$$

To alleviate the computational burden, Del Moral et al. [14] show that the joint representation of the proposal distribution $\pi_t(\boldsymbol{\xi}_{0:t})$ can be efficiently calculated as it only requires solving the product over $t$ transitions

$$\pi_t(\boldsymbol{\xi}_{0:t}) = \pi_0(\boldsymbol{\xi}_0) \prod_{\tau=0}^t K_{t+1}(\boldsymbol{\xi}_t, \boldsymbol{\xi}_{t+1}). \tag{27}$$

We define the joint proposal distribution as the empirical distribution $\pi_t(\boldsymbol{\xi}_{0:t}) = M^{-1} \sum_{i=1}^M \delta_{\boldsymbol{\xi}_{0:t}^{(i)}}(\boldsymbol{\xi}_{0:t})$ defined by a set of joint particles $\boldsymbol{\xi}_{0:t}^{(i)}$. Thus, the joint posterior approxi-

mation of $p_{\beta_t}(\boldsymbol{\xi}_{0:t}|\mathbf{x}_{1:N}^\star)$ based on the importance weights reads as

$$q_t(\boldsymbol{\xi}_{0:t}) = \sum_{i=1}^M w_t^{(i)} \delta_{\boldsymbol{\xi}_{0:t}^{(i)}}(\boldsymbol{\xi}_{0:t}); \quad w_t^{(i)} = \frac{p_{\beta_t}(\boldsymbol{\xi}_{0:t}^{(i)}|\mathbf{x}_{1:N}^\star)}{\pi_t(\boldsymbol{\xi}_{0:t}^{(i)})} \tag{28}$$

$$\Rightarrow \quad q_t(\boldsymbol{\xi}_t) = \sum_{i=1}^M w_t^{(i)} \delta_{\boldsymbol{\xi}_t^{(i)}}(\boldsymbol{\xi}_t); \quad w_t^{(i)} = \frac{p_{\beta_t}(\boldsymbol{\xi}_{0:t}^{(i)}|\mathbf{x}_{1:N}^\star)}{\pi_t(\boldsymbol{\xi}_{0:t}^{(i)})}. \tag{29}$$

The marginal target posterior approximation $q_t(\boldsymbol{\xi}_t)$ can be directly recovered from the joint approximation $q_t(\boldsymbol{\xi}_{0:t})$. Furthermore, both distributions share their weights which means that it is only required to estimate the weights $w_t$ in order to approximate the target posteriors $p_{\beta_t}(\boldsymbol{\xi}_t|\mathbf{x}_{1:N}^\star)$. In general, the probability of the target joint posterior $p_{\beta_t}(\boldsymbol{\xi}_{0:t}^{(i)}|\boldsymbol{\xi})$ is intractable. Therefore, the authors introduce an auxiliary backward Markov kernel $L_t(\boldsymbol{\xi}_{t+1}, \boldsymbol{\xi}_t)$ to simplify the computation

$$p_{\beta_t}(\boldsymbol{\xi}_{0:t}|\mathbf{x}_{1:N}^\star) = p_{\beta_t}(\boldsymbol{\xi}_t|\mathbf{x}_{1:N}^\star) \prod_{\tau=0}^{t-1} L_\tau(\boldsymbol{\xi}_{\tau+1}, \boldsymbol{\xi}_\tau). \tag{30}$$

Assuming that a posterior approximation of the target posterior $p_{\beta_t}(\boldsymbol{\xi}_t|\mathbf{x}_{1:N}^\star)$ is available through the set of weighted particles $\{(w_t^{(i)}, \boldsymbol{\xi}_t^{(i)})\}$ and the particles of the proposal distribution $\pi_t(\boldsymbol{\xi}) = M^{-1} \sum_{i=1}^M \delta_{\boldsymbol{\xi}_t^{(i)}}(\boldsymbol{\xi})$ are updated based on a kernel transition $\boldsymbol{\xi}_{t+1}^{(i)} \sim K_{t+1}(\boldsymbol{\xi}_t^{(i)}, \boldsymbol{\xi}_{t+1}^{(i)})$, then the importance weights $w_{t+1}$ are updated based on the following recursion

$$w_{t+1} = \frac{p_{\beta_{t+1}}(\boldsymbol{\xi}_{0:t+1})}{\pi_{t+1}(\boldsymbol{\xi}_{0:t+1})} = \frac{p_{\beta_{t+1}}(\boldsymbol{\xi}_{t+1}) \prod_{\tau=0}^t L_t(\boldsymbol{\xi}_{\tau+1}, \boldsymbol{\xi}_\tau)}{\pi_0(\boldsymbol{\xi}_0) \prod_{\tau=0}^t K_t(\boldsymbol{\xi}_\tau, \boldsymbol{\xi}_{\tau+1})} \tag{31}$$

$$= \underbrace{\frac{p_{\beta_{t+1}}(\boldsymbol{\xi}_{t+1}) L_t(\boldsymbol{\xi}_{t+1}, \boldsymbol{\xi}_t)}{p_{\beta_t}(\boldsymbol{\xi}_t) K_{t+1}(\boldsymbol{\xi}_t, \boldsymbol{\xi}_{t+1})}}_{\hat{w}_{t+1}} \underbrace{\frac{p_{\beta_t}(\boldsymbol{\xi}_t) \prod_{\tau=0}^{t-1} L_t(\boldsymbol{\xi}_{\tau+1}, \boldsymbol{\xi}_\tau)}{\pi_0(\boldsymbol{\xi}_0) \prod_{\tau=0}^{t-1} K_t(\boldsymbol{\xi}_\tau, \boldsymbol{\xi}_{\tau+1})}}_{w_t}. \tag{32}$$

Thus, the sequential update is performed by updating the current weights $w_t$ with the marginal weights $\hat{w}_{t+1}$. Up to now, the choice of the backward kernel has been neglected. As it is an auxiliary quantity, several approximations can be made to model the backward kernel. Del Moral et al. [14] refer to the optimal backward kernel as the Markov kernel that minimizes the variance of the particles

$$L_t^{\text{opt}}(\boldsymbol{\xi}_{t+1}, \boldsymbol{\xi}_t) = \frac{\pi(\boldsymbol{\xi}_t) K_{t+1}(\boldsymbol{\xi}_t, \boldsymbol{\xi}_{t+1})}{\pi_{t+1}(\boldsymbol{\xi}_{t+1})}.$$

They further show that the optimal backward kernel recovers the marginal weights from (25)

$$w_{t+1}^{\text{opt}} := \frac{p_{\beta_{t+1}}(\boldsymbol{\xi}|\mathbf{x}_{1:N}^\star)}{\pi_{t+1}(\boldsymbol{\xi})} = W_{t+1}. \tag{33}$$

In general, the optimal backward kernel is numerically intractable and has led to several other approximations summarized in Table B.2. Depending on choice of approximation, a number of different SMC-ABC methods have evolved, namely the classical SMC-ABC approach by Del Moral et al. [15], Population Monte Carlo (PMC)-ABC [51, 2, 31], and Metropolis-Hastings ABC [30]. Please refer to those references as well as Algorithms 2 and 3 for implementation details of these approaches.

**Algorithm 2** Sequential Monte Carlo ABC [15]

1: **input:** reference data $\mathbf{x}^\star_{1:N}$, prior $p(\boldsymbol{\xi})$, stochastic simulator $p(\mathbf{x}|\boldsymbol{\xi})$, IPM $D(\cdot,\cdot)$, max. iteration count $T$, forward kernel $K(\boldsymbol{\xi}_t,\boldsymbol{\xi}_{t+1})$, resampling threshold $V$, $\alpha$
2: initialize particles $\boldsymbol{\xi}_0^{(k)} \sim p(\cdot)$
3: initialize particle weights $w_0^{(k)} = 1/K$
4: **for** $t$ in $1\!:\!T$ **do**
5:   **for** each $\boldsymbol{\xi}_{t-1}^{(k)}$ **do**
6:     simulate $\mathbf{x}_{1:M}^{(k)} = \{\mathbf{x}_m^{(k)} \sim p(\mathbf{x}|\boldsymbol{\xi}_{t-1}^{(k)})\}$
7:     compute IPM $s_{t-1}^{(k)} = D(\mathbf{x}_{1:N}^\star, \mathbf{x}_{1:M}^{(k)})$
8:   **end for**
9:   update the bandwidth $\beta_t$ by solving

$$\text{ESS}(\{w_t^{(k)}\}, \beta_t) = \alpha \cdot \text{ESS}(\{w_{t-1}^{(k)}\}, \beta_{t-1})$$

$$w_t^{(k)} \propto w_{t-1}^{(k)} \frac{\mathbb{1}_{\{s_{t-1}^{(k)} \leq \beta_t\}}}{\mathbb{1}_{\{s_{t-1}^{(k)} \leq \beta_{t-1}\}}}$$

10:   **if** $\text{ESS}(\{w_t^{(k)}\}, \beta_t) < V$ **then**
11:     resample K particles $\boldsymbol{\xi}_t^{(k)}$ from $\{\boldsymbol{\xi}_{t-1}^{(k)}\}$
12:     set weights $w_t^{(k)} = 1/K$
13:   **end if**
14:   sample $K$ particles $\boldsymbol{\xi}_t^{(k)} \sim K(\boldsymbol{\xi}_{t-1}^{(k)}, \boldsymbol{\xi}_t^{(k)})$
15: **end for**
16: **output:** posterior particles $\boldsymbol{\xi}_T^{(k)}$

**Algorithm 3** PMC-ABC [31]

1: **input:** reference data $\mathbf{x}^\star_{1:N}$, prior $p(\boldsymbol{\xi})$, stochastic simulator $p(\mathbf{x}|\boldsymbol{\xi})$, IPM $D(\cdot,\cdot)$, max. iteration count $T$, forward kernel $K_{t+1}(\boldsymbol{\xi}_t,\boldsymbol{\xi}_{t+1})$, $\alpha$-Quantile $\alpha$
2: initialize particles $\boldsymbol{\xi}_0^{(k)} \sim p(\cdot)$
3: initialize particle weights $w_0^{(k)} = 1/K$
4: store number of best particles $K_\alpha = \alpha K$,
5: **for** $t$ in $1\!:\!T$ **do**
6:   elect $K_\alpha$ best particles $\hat{\boldsymbol{\xi}}_{t-1}^{(k)}$
7:   sample $K - K_\alpha$ proposal particles $\tilde{\boldsymbol{\xi}}_t^{(l)} \sim K_t(\hat{\boldsymbol{\xi}}_{t-1}^{(k)}, \tilde{\boldsymbol{\xi}}_t^{(l)})$
8:   **for** each $\tilde{\boldsymbol{\xi}}_t^{(l)}$ **do**
9:     simulate $\mathbf{x}_{1:M}^{(l)} = \{\mathbf{x}_m^{(l)} \sim p(\mathbf{x}|\tilde{\boldsymbol{\xi}}_t^{(l)})\}$
10:     compute IPM $s_t^{(l)} = D(\mathbf{x}_{1:N}^\star, \mathbf{x}_{1:M}^{(l)})$
11:   **end for**
12:   update bandwidth based on the empirical $\alpha$-Quantile $\beta_t = \mathcal{Q}_{\{s_{t-1}^{(k)}, s_t^{(l)}\}}(\alpha)$
13:   update weights

$$w_t^{(k)} = \frac{p(\tilde{\boldsymbol{\xi}}_t^{(k)})}{\sum_{i=1}^{K_\alpha} \frac{w_{t-1}^{(i)}}{\sum_{j=1}^{K_\alpha} w_{t-1}^{(j)}} K_t(\boldsymbol{\xi}_{t-1}^{(i)}, \tilde{\boldsymbol{\xi}}_t^{(k)})}$$

14:   set $K$ new particles $\boldsymbol{\xi}_t^k = \{\hat{\boldsymbol{\xi}}_{t-1}^{(k)}, \tilde{\boldsymbol{\xi}}_t^{(k)}\}$
15:   update forward kernel $K_{t+1}(\boldsymbol{\xi}_t, \boldsymbol{\xi}_{t+1})$
16: **end for**
17: **output:** posterior particles $\boldsymbol{\xi}_T^{(k)}$

# C Experimental details

Here we detail the experimental configurations to reproduce the results covered in Figure 3 and C.1. A small grid search has been carried out over the *learning rate*, the *trust-region parameter $\varepsilon$*, the *batch size*, and the *number of training samples* on the SLCP and Furuta task for $N = 50$. The best-fitting hyperparameters over the two tasks are reported in Table C.3 and used throughout the experiments. The remaining parameters are taken from [32] to make the results comparable.

We complement Figure 3 with Figure C.1 to compare the results in the observation space and include the Furuta pendulum.

Table B.2: Approximations of the optimal backward kernel $L_t^{\text{opt}}(\boldsymbol{\xi}_{t+1}, \boldsymbol{\xi}_t)$ lead to different SMC-ABC approaches.

| Algorithm | Assumption | $\tilde{L}_t$ | $\hat{w}_{t+1}$ |
|---|---|---|---|
| Optimal | - | $\frac{\pi_t(\boldsymbol{\xi}_t)K_{t+1}(\boldsymbol{\xi}_t,\boldsymbol{\xi}_{t+1})}{\pi_{t+1}(\boldsymbol{\xi}_{t+1})}$ | - |
| PMC-ABC | $\pi \approx p_{\beta_t}$ | $\frac{p_{\beta_t}(\boldsymbol{\xi}_t\|\mathbf{x}_{1:N}^\star)K_{t+1}(\boldsymbol{\xi}_t,\boldsymbol{\xi}_{t+1})}{\int p_{\beta_t}(\boldsymbol{\xi}_t\|\mathbf{x}_{1:N}^\star)K_{t+1}(\boldsymbol{\xi}_t,\boldsymbol{\xi}_{t+1})d\boldsymbol{\xi}_t}$ | $\frac{p_{\beta_{t+1}}(\boldsymbol{\xi}_{t+1}\|\mathbf{x}_{1:N}^\star)}{\int p_{\beta_t}(\boldsymbol{\xi}_t\|\mathbf{x}_{1:N}^\star)K_{t+1}(\boldsymbol{\xi}_t,\boldsymbol{\xi}_{t+1})d\boldsymbol{\xi}_t}$ |
| SMC-ABC | $p_{\beta_{t+1}} \approx p_{\beta_t}$ | $\frac{p_{\beta_t}(\boldsymbol{\xi}_t\|\mathbf{x}_{1:N}^\star)K_{t+1}(\boldsymbol{\xi}_t,\boldsymbol{\xi}_{t+1})}{p_{\beta_t}(\boldsymbol{\xi}_{t+1}\|\mathbf{x}_{1:N}^\star)}$ | $\frac{p_{\beta_{t+1}}(\boldsymbol{\xi}_{t+1}\|\mathbf{x}_{1:N}^\star)}{p_{\beta_t}(\boldsymbol{\xi}_t\|\mathbf{x}_{1:N}^\star)}$ |
| MH-ABC | $L_t(\boldsymbol{\xi}_{t+1}, \boldsymbol{\xi}_t) = K_{t+1}(\boldsymbol{\xi}_{t+1}, \boldsymbol{\xi}_t)$ | | $\frac{p_{\beta_{t+1}}(\boldsymbol{\xi}_{t+1}\|\mathbf{x}_{1:N}^\star)K_{t+1}(\boldsymbol{\xi}_{t+1},\boldsymbol{\xi}_t)}{p_{\beta_t}(\boldsymbol{\xi}_t\|\mathbf{x}_{1:N}^\star)K_{t+1}(\boldsymbol{\xi}_t,\boldsymbol{\xi}_{t+1})}$ |

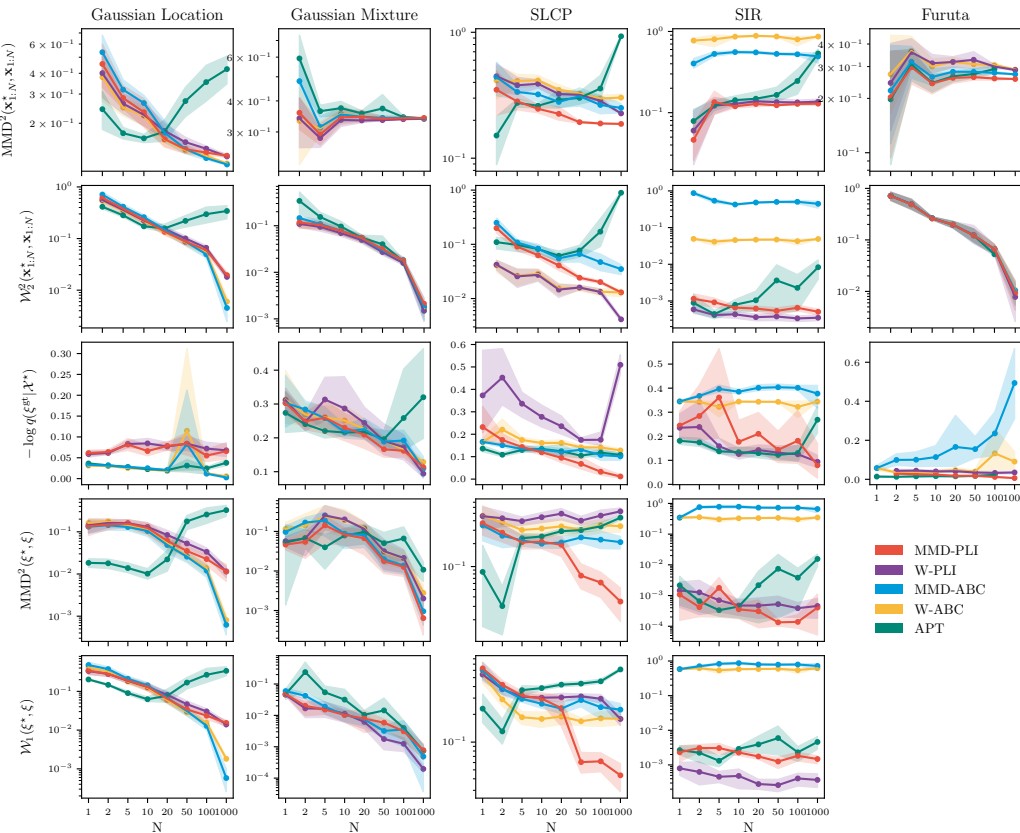

Figure C.1: Evaluation of the posterior performance on three different tasks (displayed along the columns). A node represents the mean and standard deviation of 10 experiments with different random seeds, each carried out using $N$ data points for conditioning. The samples from the approximate posterior $\boldsymbol{\xi} \sim q(\boldsymbol{\xi}|\mathbf{x}_{1:N}^\star)$ are compared against the reference posterior samples with the Wasserstein distance and the MMD when available. Additionally, the log probability of the ground-truth parameter $\boldsymbol{\xi}^{gt}$ is evaluated and posterior predictive checks are carried out on all tasks. The ground-truth parameters are described in Appendix C. Lower values are better for all metrics.

## C.1 Approximation of the integral probability metrics

In the context of this paper, we consider two instances of IPMs $D(p^\star(\mathbf{x}), p(\mathbf{x}|\boldsymbol{\xi}))$ between the data generating distribution $p^\star(\mathbf{x})$ and the likelihood $p(\mathbf{x}|\boldsymbol{\xi})$ — the maximum mean discrepancy MMD and the squared 2-Wasserstein distance $\mathcal{W}_2$. To simplify the notation, we formulate the discrepancy between the pdfs $p(\mathbf{x})$ and $q(\mathbf{x})$ whose empirical probability distributions are denoted by $\tilde{p}(\mathbf{x}) = 1/N \sum_{i=1}^N \delta_{\mathbf{x}_i}(\mathbf{x})$ and $\tilde{q}(\mathbf{y}) = 1/M \sum_{j=1}^M \delta_{\mathbf{y}_j}(\mathbf{y})$. Furthermore, we denote the cost between individual samples by $c(\mathbf{x}, \mathbf{y})$.

**Maximum mean discrepancy**   The MMD [25] can be formulated with respect to an evaluation kernel $k(\mathbf{x}, \mathbf{y})$ as the sum of three terms

$$\text{MMD}^2(p, q) = \underset{\substack{\mathbf{x} \sim p(\mathbf{x}) \\ \mathbf{y} \sim p(\mathbf{y})}}{\mathbb{E}} [k(\mathbf{x}, \mathbf{y})] - 2 \underset{\substack{\mathbf{x} \sim p(\mathbf{x}) \\ \mathbf{y} \sim q(\mathbf{y})}}{\mathbb{E}} [k(\mathbf{x}, \mathbf{y})] + \underset{\substack{\mathbf{x} \sim q(\mathbf{x}) \\ \mathbf{y} \sim q(\mathbf{y})}}{\mathbb{E}} [k(\mathbf{x}, \mathbf{y})] . \tag{34}$$

As the expectations are generally intractable, an unbiased estimate of MMD based on samples drawn from $p$ and $q$ is used [25]

$$\text{MMD}^2(\tilde{p}, \tilde{q}) \approx \frac{1}{N(N-1)} \sum_{i \neq i'}^N k(\mathbf{x}_i, \mathbf{x}_{i'}) - \frac{2}{NM} \sum_{i,j=1}^{N,M} k(\mathbf{x}_i, \mathbf{y}_j) + \frac{1}{M(M-1)} \sum_{j \neq j'}^M k(\mathbf{x}_j, \mathbf{x}_{j'}) . \tag{35}$$

Here, $\mathbf{x}_i$ and $\mathbf{y}_j$ represent samples drawn from the sampling distributions $\mathbf{x}_{1:N} \sim p(\mathbf{x})$ and $\mathbf{y}_{1:M} \sim q(\mathbf{y})$. In this paper, a Gaussian kernel $\exp(-1/(2\ell)c(\mathbf{x},\mathbf{y}))$ with bandwidth $\ell$ is employed. The bandwidth is known to be very sensitive, which is why the kernel is evaluated over a variety of bandwidths $\ell = \{1, 10, 20, 40, 80, 100, 130, 200, 400, 800, 1000\}$ by summing over the bandwidths $k(\mathbf{x}_i, \mathbf{y}_j) = \sum_\ell k_\ell(x_i, y_j)$ [16].

**Wasserstein distance.** In the experiments, we consider the squared 2-Wasserstein, which can be formulated as the solution to the optimal transport problem

$$\mathcal{W}_2^2(p, q) = \inf_\gamma \int c(\mathbf{x}, \mathbf{y}) \gamma(\mathbf{x}, \mathbf{y}) \, \mathrm{d}\mathbf{x}\mathrm{d}\mathbf{y}. \tag{36}$$

The problem is also known as the Kantorovich problem, searching for the optimal coupling $\gamma \in \Gamma(p, q)$ in the set of joint distributions that admit $p$ and $q$ through marginalization. For the empirical measures $\tilde{p}$ and $\tilde{q}$, the Kantorovich problem can be formulated as a linear program

$$\mathcal{W}_1 = \min_{\mathbf{P} \in \mathbf{U}} \sum_{i,j} P_{ij} C_{ij}. \tag{37}$$

Here, $C_{ij} = c(\mathbf{x}_i, \mathbf{y}_j)$ is the cost matrix containing the pairwise comparisons between the samples drawn from $\tilde{p}(\mathbf{x})$ and $\tilde{q}(\mathbf{y})$. The linear program searches for the optimal coupling matrix $\mathbf{P}$ among the set of doubly stochastic matrices $\mathbf{U} = \{\mathbf{P} \in \mathbb{R}_+^{N \times M} : \mathbf{P1}_M = 1/N\mathbf{1}_N, \mathbf{P}^\intercal\mathbf{1}_N = 1/M\mathbf{1}_M\}$. Peyré and Cuturi [44] show that introducing an entropy regularization term $\varepsilon H(\mathbf{P})$ to the objective (37) leads to an iterative scheme that can be solved with the Sinkhorn algorithm [46]. The iterative procedure enables parallelization on hardware accelerators to efficiently solve the optimal transport problem. Furthermore, it can be seen that the 2-Wasserstein distance can be recovered in the limit $\varepsilon \to 0$. We leverage the JAX library, OTT [9], to approximate $\mathcal{W}_2^2$ with Sinkhorn iterations for the computations.

## C.2 Gaussian location

The Gaussian location model is a 10-dimensional Gaussian model. The ten dimensional parameters $\boldsymbol{\xi} \in [-1, 1]^{10}$ define the means of the model $\mathcal{N}(\mathbf{x}|\boldsymbol{\mu} = \boldsymbol{\xi}, \boldsymbol{\Sigma} = 0.1\mathbf{I})$. We choose a Gaussian prior $p(\boldsymbol{\xi}) = \mathcal{N}(\boldsymbol{\xi}|\mathbf{0}, 0.1\mathbf{I})$ for which the posterior can be recovered in closed form. The ground-truth parameter is sampled uniformly within the posterior support $\boldsymbol{\xi}^{\mathrm{gt}} \sim \mathcal{U}(-\mathbf{1}, \mathbf{1})$.

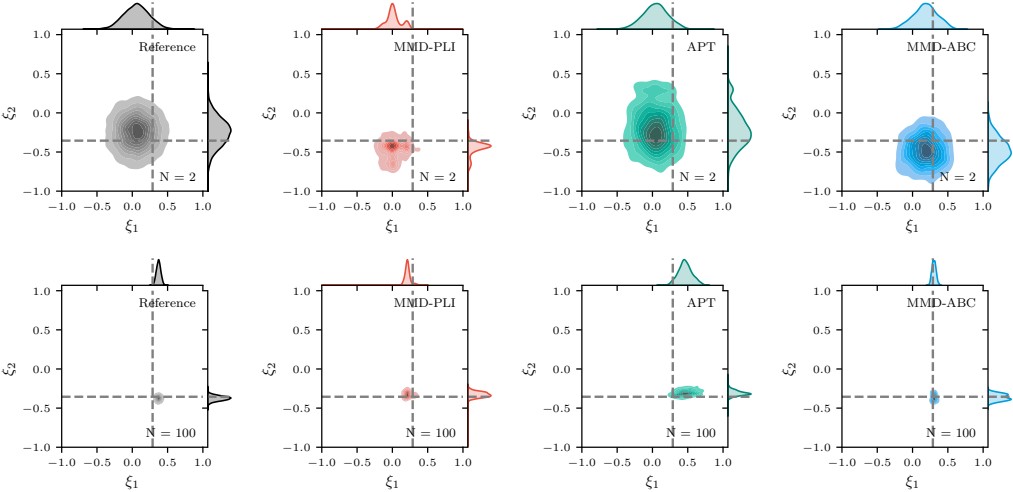

Figure C.2: Slice of the posterior through the $\boldsymbol{\xi}_1 - \boldsymbol{\xi}_2$ plane. The upper row shows experiments conducted on $N = 2$ reference observations. The lower row shows the approximate posteriors for $N = 100$ reference observations. The dotted line represents the ground truth parameter $\boldsymbol{\xi}^{(\mathrm{gt})}$ that was used to generate $\mathbf{x}_{1:N}^\star$. All approaches show that the posterior becomes denser when conditioned on more data.

## C.3 Gaussian mixture model

The task is to infer the mean parameters of a two-dimensional multivariate Gaussian mixture model [47]

$$p(\mathbf{x}|\boldsymbol{\xi}) = \mathcal{N}(\mathbf{x}|\boldsymbol{\mu} = \boldsymbol{\xi}, \boldsymbol{\Sigma} = \mathbf{I}) + \mathcal{N}(\mathbf{x}|\boldsymbol{\mu} = \boldsymbol{\xi}, \boldsymbol{\Sigma} = 0.01\mathbf{I}) \tag{38}$$

The two-dimensional observation space represents samples from the Gaussian mixture model. We assume a uniform prior $p(\boldsymbol{\xi}) = \mathcal{U}(-\mathbf{10}, \mathbf{10})$.

## C.4 Simple-likelihood complex-posterior

The Simple-Likelihood Complex-Posterior (SLCP) task consists of a 5-dimensional parameter space $\boldsymbol{\Xi} \in [-3, 3]^5$ with a uniform prior $\mathcal{U}(-\mathbf{3}, \mathbf{3})$. The ground-truth parameter, from which the reference observations are generated, is set to $\boldsymbol{\xi}^{(\mathrm{gt})} = (0.7, 1.5, -1.0, -0.9, 0.6)^\intercal$. The observations represent four samples from a 2-dimensional Gaussian distribution

$$\mathbf{x} = [\mathbf{x}_1, \mathbf{x}_2, \mathbf{x}_3, \mathbf{x}_4,]^\intercal, \quad \mathbf{x}_i = \mathcal{N}(\mathbf{x}_i, \boldsymbol{\mu}(\boldsymbol{\xi}), \boldsymbol{\Sigma}(\boldsymbol{\xi})). \tag{39}$$

For further information, we refer to the SBI benchmarking paper from Lueckmann et al. [32].

## C.5 SIR

The SIR model is an epidemiological time-series model that models the spreading of a disease. The name derives from the three states, (i) susceptible, (ii) infectious, (iii) recovered, that an individual can be in. The parameters of the dynamics model are the contact rate $\beta$ and the mean recovery rate $\gamma$

$$\boldsymbol{\xi} = \begin{bmatrix} \beta \\ \gamma \end{bmatrix} \in \begin{bmatrix} (0, 2] \\ (0, 0.5] \end{bmatrix}. \tag{40}$$

The prior is a log-normal distribution over $\beta$ and $\gamma$

$$\beta \sim \mathrm{LogNormal}(\log(0.4), 0.5) \tag{41}$$

$$\gamma \sim \mathrm{LogNormal}(\log(0.125), 0.2) \tag{42}$$

We rollout the dynamics over 160 timesteps and evaluate the simulation at 20 equidistant time-steps by taking a sample from the binomial, $x_i \sim \mathrm{Binom}(1000, I_i/N)$. Here $N$ denotes the total population at the start of the simulation. See Lueckmann et al. [32] for details on the dynamics of the SIR model.

## C.6 The Furuta pendulum

This inverted double pendulum can be described by the angular deflection $[\theta_r, \theta_p]$ of the rods *w.r.t.* their equilibrium position $[0, 0]$. The equations of motion can be derived by formulating the Euler-Lagrange equation [37]:

$$\begin{bmatrix} -d_r\dot{\theta}_r + \tau \\ -d_p\dot{\theta}_p \end{bmatrix} = \begin{bmatrix} \frac{1}{12}m_r l_r^2 + m_p l_r^2 + \frac{1}{4}m_p l_p^2 \sin^2\theta_p & \frac{1}{2}m_p l_p l_r \cos\theta_p \\ \frac{1}{2}m_p l_p l_r \cos\theta_p & \frac{1}{3}m_p l_p^2 \end{bmatrix} \begin{bmatrix} \ddot{\theta}_r \\ \ddot{\theta}_p \end{bmatrix}$$
$$+ \begin{bmatrix} \frac{1}{4}m_p l_p^2 \sin 2\theta_p \, \dot{\theta}_r\dot{\theta}_p - \frac{1}{2}m_p l_p l_r \sin\theta_p \, \dot{\theta}_p^2 \\ -\frac{1}{8}m_p l_p^2 \sin 2\theta_p \, \dot{\theta}_r^2 + \frac{1}{2}m_p l_p g \sin\theta_p \end{bmatrix}. \tag{43}$$

Here, the mild assumption is made that the pole length is significantly greater than its diameter for which the moments of inertia of the poles around their pivot are $J_i = 1/3 \, m_i l_i^2$, $i \in \{r, p\}$. The mass matrix contains entries from the translatory and rotational movement of the two poles. As the reference coordinate systems are constantly rotating *w.r.t.* the basis coordinate system, Coriolis forces occur. They are complemented by gravitation which works on the rotational pole. The left-hand side considers damping in the joints, represented by the damping coefficients $d_r$ and $d_p$, and the torque $\tau$ which is applied from a servo motor. For this paper, we omit external forces, i.e., $\tau = 0\,\mathrm{Nm}$. The Furuta pendulum is set into motion by perturbing the initial state around its unstable equilibrium.

For the system identification tasks we select the five system parameters

$$\boldsymbol{\xi} = \begin{bmatrix} g \\ l_r \\ m_r \\ l_p \\ m_p \end{bmatrix} \in \begin{bmatrix} [9, 11] \\ [0.08, 0.09] \\ [0.08, 0.1] \\ [0.12, 0.135] \\ [0.02, 0.03] \end{bmatrix}; \quad \boldsymbol{\xi}^{\mathrm{gt}} = \begin{bmatrix} 9.81 \\ 0.085 \\ 0.095 \\ 0.129 \\ 0.024 \end{bmatrix}$$

with a uniform prior on the predefined ranges.

Table C.3: Hyper-parameter settings of the SBI methods as used for the experiments in Section 4. Forward slashes symbolize layers of a neural network.

| Parameter | Value |
|---|---|
| | PLI (Ours) |
| Likelihood kernel | Exponential Kernel |
| Trust-region threshold $\varepsilon$ | 0.5 |
| Model | Neural Spline Flow (NSF) |
| Bijector | Rational Quadratic Spline with param size $D$ |
| # Bins | 10 |
| Conditioning MLP | input dim / 50 / 50 / 50 / $D$ |
| # Bijectors / Transforms | 5 |
| Base distribution | $\mathcal{N}(\mathbf{0}, \mathbf{1})$ |
| Learning rate | $1 \times 10^{-5}$ |
| Epochs | 20 |
| Train samples per iteration | 5000 |
| Batch size | 125 |
| | PMC-ABC [31] |
| Likelihood kernel | Uniform Kernel |
| Likelihood update | $\alpha$-Quantile, $\alpha = 0.1$ (see Table A.1) |
| $\alpha$ | 0.1 |
| Reverse transition kernel $\tilde{L}_t$ | reverse PMC kernel (see Table B.2) |
| Particles | 1000 |
| Epochs | 200 |
| Perturbation kernel | GMM with 5 components |
| | APT [24] |
| Model | Conditional Neural Spline Flow (NSF) |
| Bijector | Rational Quadratic Spline with param size $D$ |
| # Bins | 10 |
| # Bijectors / Transforms | 5 |
| Conditioning MLP | input dim / 32 / 32 / 32 / $D$ |
| Base distribution | $\mathcal{N}(\mathbf{0}, \mathbf{1})$ |
| # Atoms | 10 |
| Learning rate | $1 \times 10^{-5}$ |
| Epochs | 20 |
| Train samples per epoch | 5000 |
| Batch size | 500 |

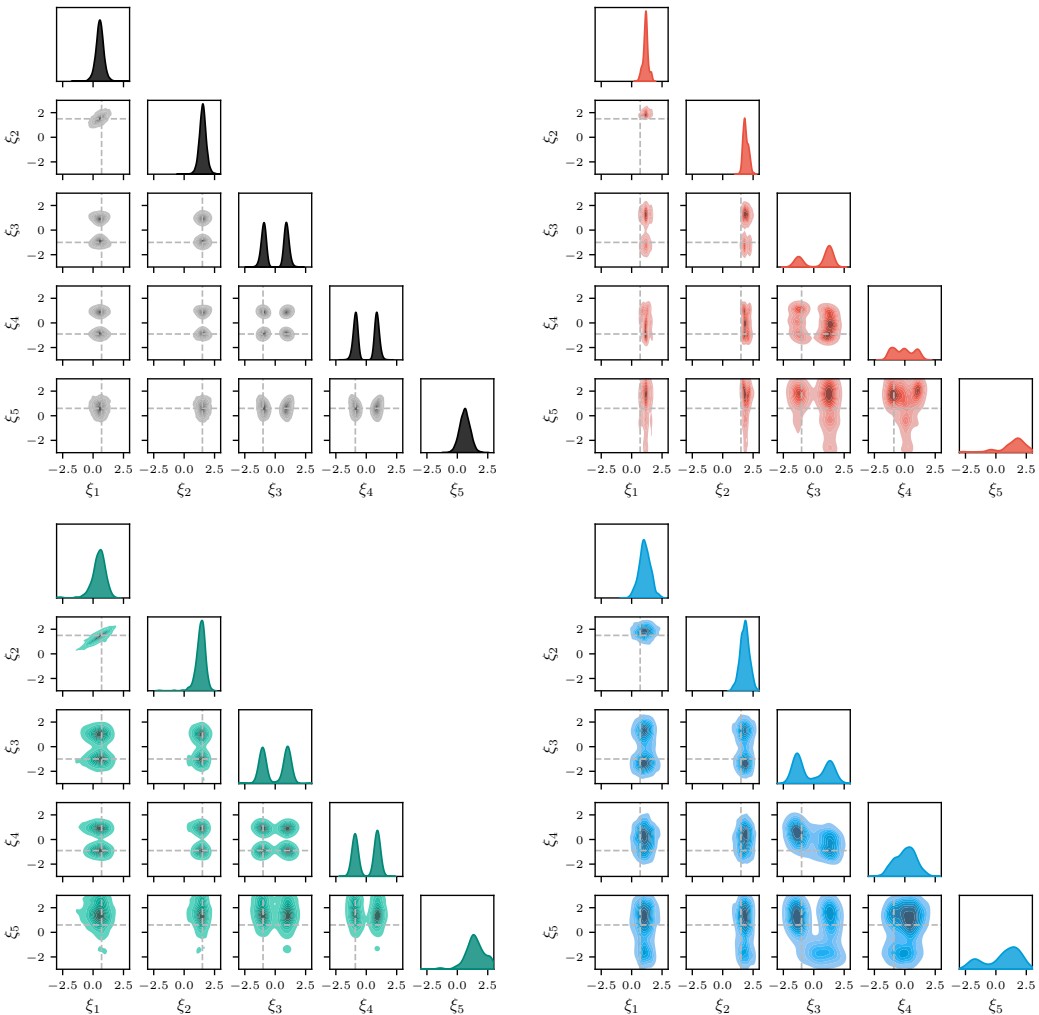

Figure C.3: Results of posterior inference on the SLCP task with $N = 2$ reference observations. The unimodal distribution of the parameters $\xi_1$ and $\xi_2$) are depicted well by all approaches. On the contrary, the multi-modality is only represented properly by the APT posterior (▬ Reference, ▬ MMD-PLI, ▬ APT, ▬ MMD-ABC).

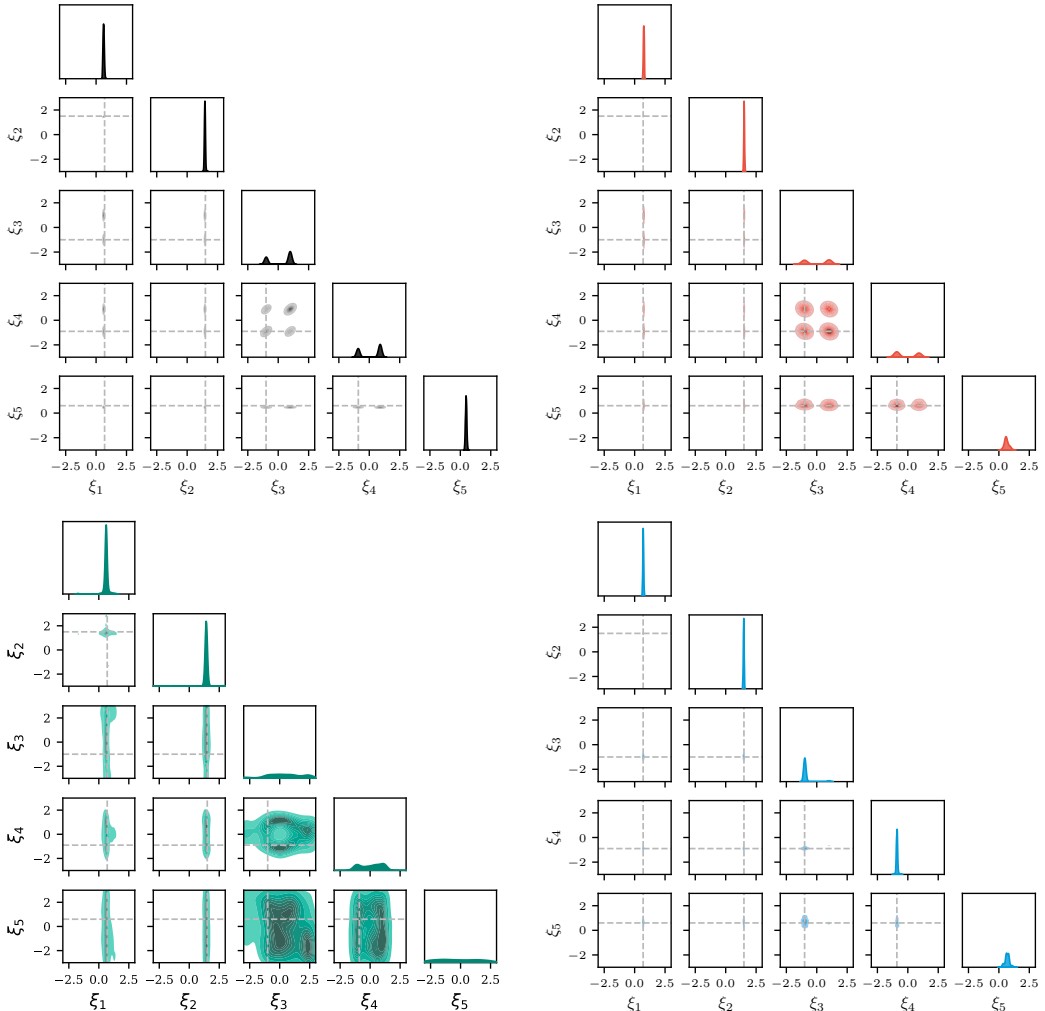

Figure C.4: Results of posterior inference on the SLCP task with $N = 100$ reference observations. Compared to the posterior given $N = 2$ observations (Figure C.3), the posterior (━ Reference) is distributed tightly around distinct points. Here, ━ MMD-PLI captures all modes of the posterior, ━ MMD-ABC centers around a uni-mode, while ━ APT cannot represent the multi-modality.

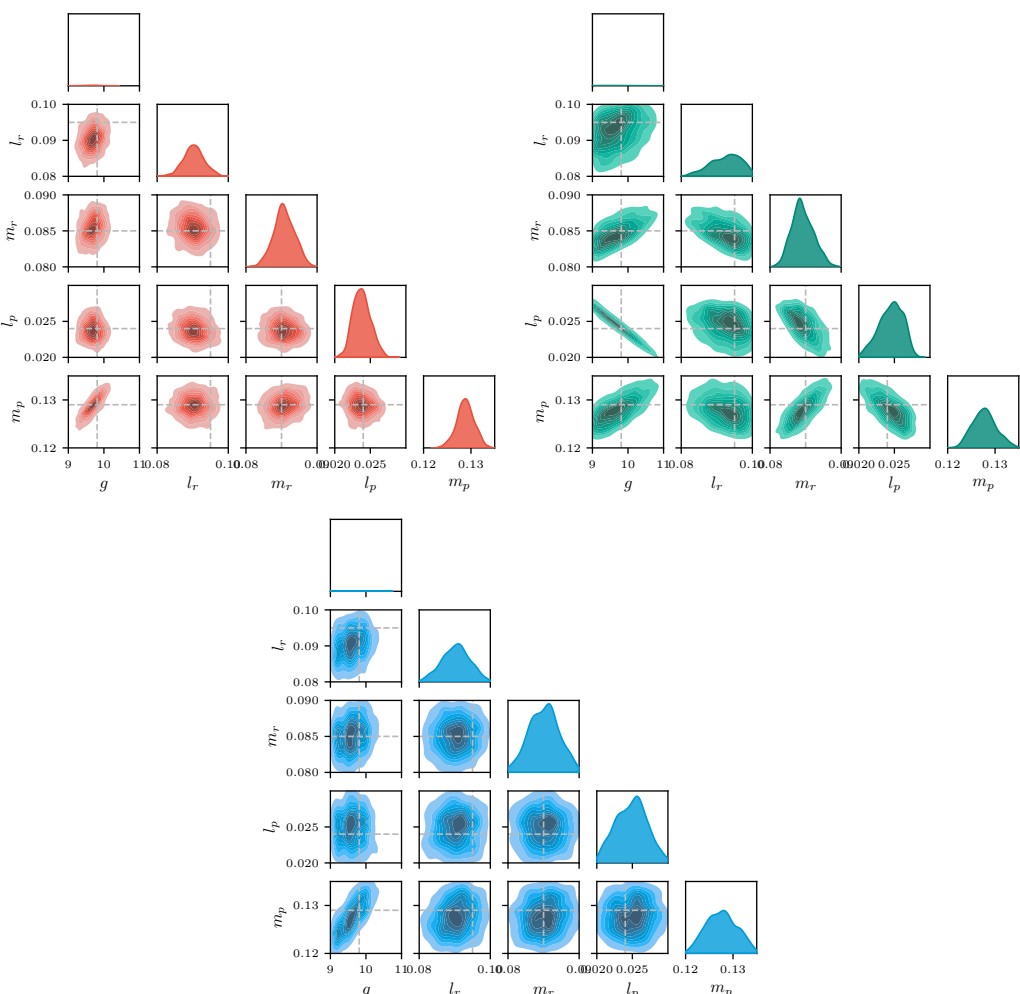

Figure C.5: Results of posterior inference on the Furuta pendulum with $N = 2$ reference observations. All methods center around the ground truth parameter. ▬ APT finds the expected correlations among the parameters while ▬ MMD-PLI and ▬ MMD-ABC remain more widespread.

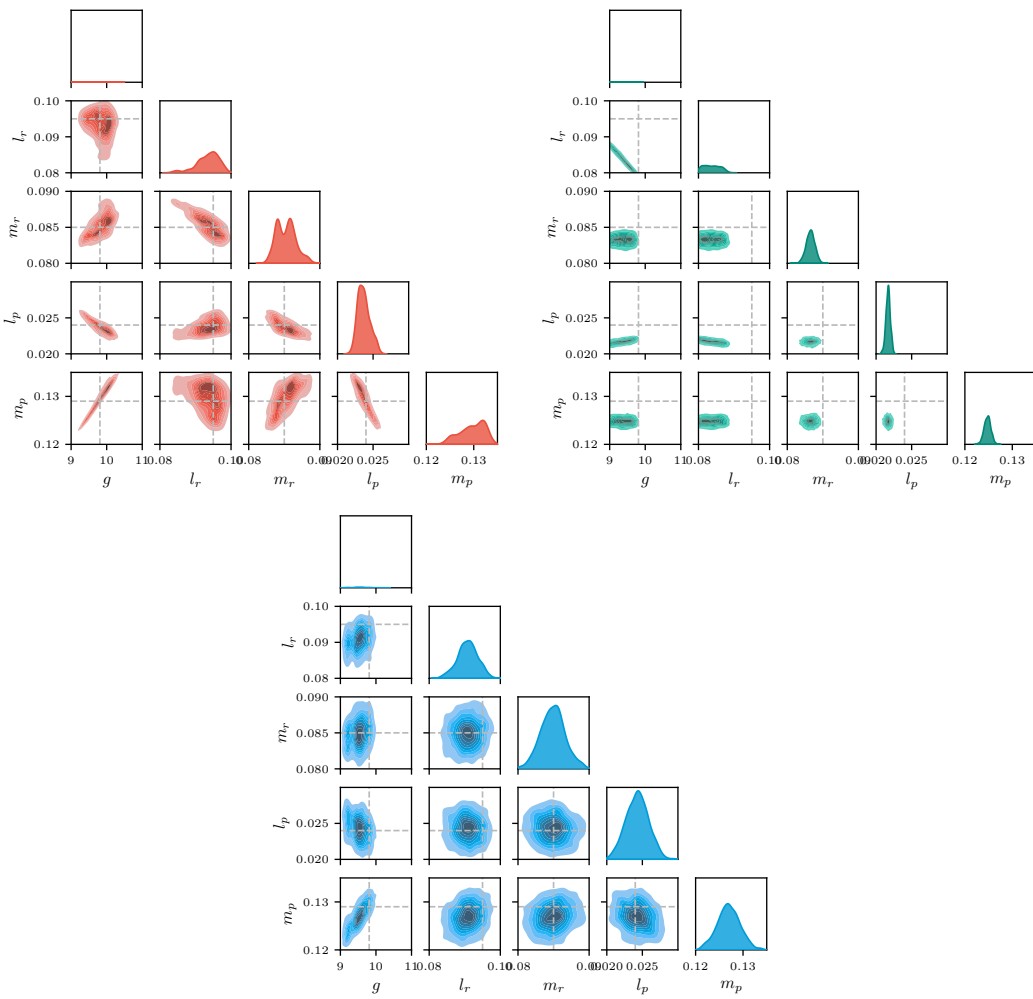

Figure C.6: Results of posterior inference on the Furuta pendulum with $N = 100$ reference observations. All models capture the ground-truth parameter well. In contrast to the $N = 2$ setting (Figure C.5) ▬ MMD-PLI reveals pairwise correlations between the domain parameters, and ▬ MMD-ABC is less densely distributed. Note, that ▬ APT clusters outside of the ground-truth parameter.

