# OpenReview forum: "Pseudo-Likelihood Inference"
_NeurIPS.cc/2023/Conference — NeurIPS 2023 poster_

### Official Review · Reviewer_7dVy · 2023-07-04

**Soundness:** 3 good
**Presentation:** 3 good
**Contribution:** 3 good
**Rating:** 6
**Confidence:** 4

**Summary:**

This method perform simulation-based inference with a hybrid of previous methods, in this case integrating a neural likelihood-type step within a SMC-ABC algorithm for performing posterior inference. It considers use of distance metrics such as the Wasserstein and the Maximum Mean Discrepancy to compare simulated and observed data, while performing particle-based updates in a fairly standard way for SMC-type methods. It performs a comparison on five different test examples, with comparisons with different ABC methods and a neural likelihood method.

**Strengths:**

The method is fairly well presented, at least the steps setting up the framework of the algorithm used.

The examples used are appropriate, albeit limited to low-dimensional parameter spaces. The Furuta pendulum, which I had not seen before, is particularly nice.

You have chosen a few competing methods which, by no means exhaustive, at least make similar assumptions to PLI and therefore the comparison is revealing.

“Related Work” is fairly thorough.


**Weaknesses:**

There are a few little typoes in the language that I noticed: "Posteriror” “numerica”. I imagine there are more that a standard spellcheck would help with.

The actual algorithm and computational steps used to estimate the relevant divergences could be much more thoroughly explained, i.e. “Compute IPM” without saying how to for MMD vs Wasserstein vs other is unclear. You provide a reference for this estimation but I think it should be better explained in this article itself.

I’m not sure that I really buy the analogy with variational inference: the proposal distribution formulation built with weights is fairly standard from the SMC literature, and I don’t think the variational angle is particularly clear. I imagine that you do have a legitimate point to make in this respect, so if you could expand your reasoning on this line then that would be a good thing.

A few specific claims that I didn't really agree with:

“There might be several parameter configurations yielding the same observation, hence rendering the resulting distribution to be multi-modal.” - Multi-modality is not always a result of non-identifiability, just different parameter values having comparable likelihoods, not necessarily the same observed data.

“KL divergence is intractable in SBI because we cannot evaluate the likelihood” - can be evaluated without the likelihood with density ratio estimation and classifiers as in classifier ABC and f-divergences. FIND CITATIONS.

The term “likelihood kernel” is introduced without explanation.




**Questions:**

The Furuta pendulum is a nice example. Are there other examples of using chaotic systems rather than stochastic systems as test cases for simulator based inference? Do you treat the variation between two adjacent deterministic chaotic systems as being basically random, and is there a downside to making that assumption?

Is “APT” in the results section basically targeting the posterior directly without the SMC-type exploration? Why does it seem to do well up to a given value of N then fall off quite extremely? Is a more concentrated posterior (larger N) more difficult to explore, and the SMC-type method helps with exploration of a complex posterior? Constructing a neural likelihood approximation at every iteration must add quite a large computational overhead that should be acknowledged.

What is the relevance of model misspecification here? This is acknowledged in the section describing General Variational Inference, and it would be nice to see at least a small discussion of the possible advantages of using the MMD or Wasserstein compared to the KL for the “true” likelihood.

What is the expected scaling with dimensionality of the parameter space? You have constrained yourself to low-dimensional examples in this paper. The density-ratio type methods tend to do quite well in higher-dimensional spaces as neural nets can fit in this spaces with enough data, but SMC suffers quite severely with the curse of dimensionality. Does the use of neural networks in PLI help overcome this curse? If yes, great! If no, is it worth the overhead of training a neural net in a lower-dimensional space when there might exist other methods that do the job adequately? Would the straight APT method do better than any of the SMC-type methods in a higher-dimensional space?


**Limitations:**

No clear immediate social impact of this work. It really depends on the model on which inference is being performed.

---

> ### Author Rebuttal · Authors · 2023-08-09
>
> ### Answer to Reviewer 7dVy
> We thank the reviewer for the extensive feedback.
>
> **Remarks and Questions:**
> > [explain] computational steps used to estimate the relevant divergences.
>
> We will add:
> *We follow [1] to approximate the MMD for two discrete probability measures, whereas entropy-regularized optimal transport is used to approximate the Wasserstein distance [2]. Both versions facilitate parallelization on the GPU.*
>
> And we added a subsection to Appendix C describing the exact implementation of MMD and the Wasserstein distance.
>
> > [...] the variational angle is [not] clear.
>
> Viewing PLI from either the SMC or the VI perspective makes the method interesting. At each inference iteration, we solve the variational inference problem in eq. (6). The closed-form solution of the constrained optimization problem results in a weighted formulation which shares many characteristics with SMC methods. Thus, our method is similar to Particle Mirror Descent [3] and Particle-Based Variational Inference [4] in that they also represent the variational distribution using particles. Therefore, we also consider the particle-based representation as a type of variational family.
>
> > Multi-modality is not always a result of non-identifiability [...]
>
> Though our statement does not exclude the case of having different observations with the same likelihood, we agree that we should rephrase it for completeness:
> *Additionally, several parameter configurations may yield observations with comparable likelihood, rendering the posterior distribution multi-modal.*
>
> > [KL] can be evaluated without the likelihood [...]
>
> To be precise, we indeed cannot "evaluate" the likelihood, but of course, if the KL divergence is approximated with another estimator, then we could use it. This might be an interesting idea to explore in future work. As we intended to outline that the KL cannot be estimated without specifying an adversarial model, we update the formulation accordingly:
> *Without specifying additional estimators, the KL is intractable in SBI, ...*
>
> > The term “likelihood kernel” is introduced without explanation.
>
> The likelihood kernels are presented in Table A.1. in the appendix. In the ABC setting, the likelihood kernel represents the uniform kernel with rejection threshold $\beta$, whereas the likelihood kernel in PLI refers to the exponential kernel in eq.(6). We will ensure it is well defined in the final version.
>
> > Are there other examples of using chaotic systems [...]
>
> We are unaware of examples of using chaotic systems in the SBI literature. This seems like an exciting direction for investigation.
>
> > Do you treat the variation between two adjacent deterministic chaotic systems as being basically random, and is there a downside to making that assumption?
>
> We did not make this assumption explicitly. But it is a very insightful question how and if such assumptions influence the applicability of SBI methods for deterministic chaotic systems.
>
> > Is “APT” in the results section basically targeting the posterior directly without the SMC-type exploration?
>
> Indeed "APT" is directly targeting the posterior distribution during training. The method leverages a proposal distribution and the simulation data to construct a training dataset based on maximum likelihood estimation to train the posterior. The APT formulation allows bootstrapping the posterior of previous iterations as a new sampling distribution, similar to PLI, and accounting for it through importance reweighting.
>
> > Why does [APT] fall off quite extremely? [...]
>
> As APT is solely trained on tuples of parameters and simulations, higher dimensional problems that include larger N might require more data or an inductive bias, as introduced in PLI through the weights.
>
> > What is the relevance of model misspecification here?
>
> Model misspecification denotes the discrepancy between the data generating distribution $p^\star(x)$ of the data generating process and the simulated samples $p(x|\xi)$. If the data generating model contains outliers, evaluation of $\text{KL}(p^\star(x)|| p(x|\xi))$ for such an outlier would lead to large weights, which heavily influence inference [5]. [6] and [7] show that MMD is more robust to outliers as the MMD posterior concentrates tightly around the data generating distribution at an $\sqrt{N}$-rate
>
> $$\lim_{N\rightarrow \infty}E_{\xi \sim \pi(\xi)}[\text{MMD}(p^\star(x), p(x|\xi))] \leq \text{inf}_\xi \text{MMD}(p^\star(x), p(x|\xi)).$$
>
> The robustness property is interesting in real-world scenarios where the data generation is often accompanied by sensory noise, and models of physical systems can only be accurate to a certain degree.
>
> > What is the expected scaling with dimensionality of the parameter space? You have constrained yourself to low-dimensional examples in this paper.
>
> We chose those experiments because they are prevalent benchmarking tasks in the related literature. Further, we did introduce the Furuta benchmark to show the scalability. This paper aims to introduce and justify the presented PLI approach. For future work, we agree that scaling PLI to higher-dimensional problems is exciting and should outline the strength of the methodology even more.
>
> > Would the straight APT method do better than any of the SMC-type methods in a higher-dimensional space?
>
> We expect PLI and APT to scale better to high-dimensional problems than the ABC approaches. We believe that the weighted maximum likelihood approach adopted in PLI should make the optimization of the posterior more robust than APT, especially when scaling to higher dimensions. As APT is only trained on pairs of parameters and observations, we suspect that the additional inductive bias of PLI should be beneficial in guiding the training.
>
> [1] Cuturi (2013)
>
> [2] Gretton et al. (2012)
>
> [3] Dai et al. (2016)
>
> [4] Liu et al. (2019)
>
> [5] Knoblauch et al. (2019)
>
> [6] Chérief-Abdellatif et al. (2020)
>
> [7] Dellaporta et al. (2022)

---

> > ### Comment · Reviewer_7dVy · 2023-08-14
> >
> > Thank you for the informative rebuttal.
> >
> > Regarding the APT falling off with larger N: is it really falling off with larger p, and we have a correlation in the examples with number of data points and dimensionality? Are there any results which can make clear that this is really the effect here, i.e. a large N small p example, or small N and large p? Is there good theoretical reasons why to expect a scaling with either N or p for APT, and can you make that explicit in the text?

---

> > > ### Author Response · Authors · 2023-08-14
> > >
> > > Thank you again for the time and investment in reviewing our proposed method. Here we attempt to answer your additional questions:
> > >
> > > There have been no comprehensive evaluations of the common SBI benchmark methods on problems with high-dimensional input spaces in the literature (i.e., for large N). Similarly, theoretical analysis of the scaling of such methods is also lacking. Regarding increasing p (which we assume you mean the number of training samples), we expect APT to perform well for high N. Nevertheless, it's important to highlight that this is accompanied by a higher sample complexity, while a primary objective of PLI is to enhance sample efficiency. Consequently, we emphasize evaluating its performance within the constraints of a small, fixed simulation budget. Under these conditions, we observe that PLI surpasses APT on tasks with increased N. To better understand the scaling of PLI and APT, we agree that performing ablation on p and N would be interesting, which we will include in the final paper.

---

> > > > ### Comment · Reviewer_7dVy · 2023-08-16
> > > >
> > > > I actually mean the precise opposite of what you have assumed: using N to mean number of samples and p to mean dimensionality is totally standard within the statistics community. What do you mean in the light of this?

---

> > > > > ### Author Response · Authors · 2023-08-18
> > > > >
> > > > > Thank you for providing additional clarity. We approached the question with respect to our paper's context, where N represents the number of observations utilized for inference. In order to eliminate any potential confusion, we will adopt the suggested notation in our forthcoming response:
> > > > > - p: Dimensionality of the data, indicating the number of observations multiplied by the dimension of each observation
> > > > > - N: Count of training samples
> > > > >
> > > > > Apart from interchanging the symbols p and N, the core message of our original response remains unaltered. Kindly refer to our revised answer below:
> > > > >
> > > > > There have been no comprehensive evaluations of the common SBI benchmark methods on problems with high-dimensional input spaces in the literature (i.e., for large p). Similarly, theoretical analysis of the scaling of such methods is also lacking. Regarding increasing N, we expect APT to perform well for high p. Nevertheless, it's important to highlight that this is accompanied by a higher sample complexity, while a primary objective of PLI is to enhance sample efficiency. Consequently, we emphasize evaluating its performance within the constraints of a small, fixed simulation budget. Under these conditions, we observe that PLI surpasses APT on tasks with increased p. To better understand the scaling of PLI and APT, we agree that performing ablation on p and N would be interesting, which we will include in the final paper.

---

### Official Review · Reviewer_dHap · 2023-07-05

**Soundness:** 3 good
**Presentation:** 3 good
**Contribution:** 3 good
**Rating:** 6
**Confidence:** 3

**Summary:**

This paper proposes a Pseudo-Likelihood Inference approach that brings neural approximation into ABC. This new approach can condition on a variable number of observations unlike previous neural-based SBI approaches.

**Strengths:**


* The incorporation of Sequential Monte Carlo into SBI is a strength. SMC seems like it should be related to SBI as it follows the structure (as shown in Fig. 1) of annealing to the desired distribution.
* The literature review is a strength and highlights the authors' knowledge on the topic.
* The paper is well-written.


**Weaknesses:**

* The experiments section could be strengthened to highlight the advantage of the approach compared to SNPE. For example, is the a better way to highlight that $N \not= M$ for PLI? It seems a shame not to highlight this strength of the new approach. Also, does $N=M$ mean that for APT, when $N=1$, only one training point is used to train the normalizing flow?
* In Lueckmann et al. (2021) there are actually 10 benchmark experiements. Is there a reason that only four were used here? Additionally, if “benchmarking is not the focus” (line 247) then what is the focus? This motivation could be clearer.
* The computational cost is not highlighted in the paper. Is the cost of retraining the density estimator at each step equivalent to APT?
* It is a weakness that the code was not provided.


**Questions:**

* The hyperparameters seem to be set without mention of any hyperparameter optimization. What effect does the number of particles have and the number of steps have on the overall performance? What do bins mean on the Supplementary materials?

**Limitations:**

* The idea appears novel and is the main reason for the slightly positive score. The experimentation could definitely be improved, as well as the explanation behind various hyperparameters and what the limitation of the approach is when it comes to computational cost.

---

> ### Author Rebuttal · Authors · 2023-08-09
>
> ### Answer to Reviewer dHap
> We thank the reviewer for the insightful questions and feedback.
>
> **Remarks and Questions:**
> >  Is there a better way to highlight that $N\neq M$ for PLI?
>
> Indeed, choosing $M > N$ is a strength of PLI and ABC that we could have highlighted more. We expect both methods to be more competitive against APT when using $M>N$, particularly when only a few reference observations $N$ are available. Our reasoning for restricting the evaluation to $M=N$ was to keep the simulation budget comparable.
>
> > In Lueckmann et al. (2021) there are actually 10 benchmark experiments. Is there a reason that only four were used here? Additionally, if “benchmarking is not the focus” (line 247) then what is the focus?
>
> We intend to guide the experiments as closely as possible to Lueckmann et al.'s (2021) benchmarking paper—note, however, that there the number of observations is fixed to a single sample N=1. As the generation of the reference posterior samples with MCMC methods proved to become more unstable for larger N, we only showed the results for environments in which we found good posterior samples for up to N=1000 reference observations. For further details on the choice of environments, we refer to our general response.
>
> > The computational cost is not highlighted in the paper. Is the cost of retraining the density estimator at each step equivalent to APT?
>
> Yes, the training and simulation times of PLI and APT are equivalent. The only computational overhead of PLI is the calculation of the pseudo-likelihood, which is negligible. For a more thorough discussion of the computational effort, we refer to our general rebuttal response above.
>
> > It is a weakness that the code was not provided.
>
> We, unfortunately, did not have the code cleaned and ready to publish for the initial submission. As of now the code is clean and will be published with the final paper.
>
> Questions:
> > The hyperparameters seem to be set without mention of any hyperparameter optimization.
>
> We did perform a grid search for the learning rate, the number of training samples, the trust-region parameter $\varepsilon$, and the batch size on the SLCP and Furuta tasks for $N=50$. The remaining model-specific parameters of APT were taken from the SBI benchmark and adopted to PLI to make the methods as comparable as possible. We added a note to the experimental details section in the appendix.
>
> > What effect does the number of particles have and the number of steps have on the overall performance?
>
> One of the main difficulties of simulation-based inference is the exploration-exploitation trade-off in the parameter space. Increasing the number of particles, in general, helps exploration as the search space is better covered. The number of steps is typically coupled to the perturbation kernel. Increasing the number of steps facilitates smaller perturbations, thus enabling a more fine-grained search in the parameter space. In SMC-ABC, this quantity can be tuned by the choice of either the $\alpha$-Quantile in PMC-ABC or the effective sample-size parameter $\alpha$ and resampling threshold $V$ in SMC-ABC. Further information can be found in [1].
>
> > What do bins mean on the Supplementary materials?
>
> The number of bins is a particular parameter of neural spline flows [2], denoting the number of rational-quadratic polynomial segments. The introduction of monotonic polynomial splines has been popularized in normalizing flows that use coupling transforms [2]. Coupling transforms is a common technique to make the Jacobian determinant of the change of variables tractable.
>
> [1] Lenormand, M., Jabot, F., & Deffuant, G. (2013). Adaptive approximate Bayesian computation for complex models.
>
> [2] Durkan, C., Bekasov, A., Murray, I., & Papamakarios, G. (2019). Neural spline flows.

---

> > ### Comment · Reviewer_dHap · 2023-08-14
> > **Response**
> >
> > Thanks for the clarifications. It is still not great that the code was not provided with the submission as that is normally a key part of the work and does not help with assessing reproducibility.
> >
> > However, I will raise my score since the authors did provide a response to every weakness that I highlighted.
> >
> > Thanks for your effort.

---

> > > ### Author Response · Authors · 2023-08-18
> > >
> > > Thank you again for the time and investment in reviewing and acknowledging our proposed method.

---

### Official Review · Reviewer_9BA6 · 2023-07-06

**Soundness:** 3 good
**Presentation:** 3 good
**Contribution:** 2 fair
**Rating:** 6
**Confidence:** 3

**Summary:**

This paper proposes an approch to deal with posterior inference when the likelihood is intractable, which is often the case for problems involving simulators of physical processes.
The paper investigates Approximate Bayesian Computations (ABC) methods, and it proposes incorporating neural approximations through variational principles.
The main strenghts of the proposed approach include the possibility to carry out optimization of neural posteriors via gradient descent, the lack of reliance on summary statistics, and the possibility to handle multiple observations as input.


**Strengths:**

The paper is well written and easy to follow.

The methodological developments seem solid and well motivated.

Experiments seem to support the main message of the paper.


**Weaknesses:**

Experiments could have included a larger variety of scenarios involving other types of data and simulators.


**Questions:**

As a non-expert on ABC I found the paper interesting and well written, although I find it difficult to assess the novelty of this work.
Replacing the KL with some objective that can be estimated through samples is a good idea in general, and this paper does it in eq. 6; not sure how novel the ABC community would consider this.
Similarly, moving away from modeling the posterior through particles in favor of neural density estimators yields clear advantages, and again I'm not sure how novel this would be considered in this literature.
Having said that, I found the motivation behind the work convincing and the idea well executed with experiments showing improvements over alternative approaches from the state-of-the-art.
As a result, I have an overall positive opinion of the paper which combines sensible ideas to produce an approach which seems to work well across a number of experiments.


**Limitations:**

I haven't seen any specific text on the limitations of the approach and I think this could be included in the revision.

---

> ### Author Rebuttal · Authors · 2023-08-09
>
> ### Answer to Reviewer 9BA6
> We thank the reviewer for the valuable feedback. We appreciate the *"overall positive opinion"* on the methodology but would like to summarize the main contributions of PLI again as the reviewer *"find[s] it difficult to assess the novelty of this work"*
>
> We want to motivate the novelty of PLI from the perspective of ABC and APT. The idea of replacing summary statistics with statistical measures has been extensively studied in ABC and has been shown to improve ABC methods significantly. The PLI formulation allows leveraging statistical measures to train neural posterior models, thus making the training more robust compared to APT. In the SBI community, training neural posterior models has been typically restricted to generating a training dataset using the simulator and optimizing the neural model based on the maximum likelihood principle. The introduction of weighted maximum likelihood can be considered novel from the perspective of SBI. In short, individually, the two ideas - using statistical measures and training neural posteriors - are already known and popular in their respective fields, but our PLI algorithm brings them together in the context of ABC and thereby achieves state-of-the-art performance.
>
> **Remarks:**
> > Experiments could have included a larger variety of scenarios involving other types of data and simulators.
>
> We evaluate the PLI methodology based on common benchmarking tasks in the SBI community and extend the benchmark to scenarios where $N>1$. We agree that it is interesting to evaluate how well SBI methods scale to other types of data, in particular scaling to high dimensional problems. However, note that we are already going beyond what is commonly done in SBI papers regarding problem dimensionality. In particular, we introduced the Furuta pendulum task as a new benchmark featuring multi-modal high-dimensional observation space. As experiments with high-dimensional observation and parameter spaces are quite compute-intensive, we believe it is reasonable to consider them in future work. For additional details on the choice of evaluation tasks, please refer to our overall rebuttal response.

---

> > ### Comment · Reviewer_9BA6 · 2023-08-16
> >
> > Many thanks for your clarifications and for summarizing the novel aspects charaterizing your work - no need for further clarifications at this stage.

---

### Official Review · Reviewer_AzJv · 2023-07-06

**Soundness:** 3 good
**Presentation:** 2 fair
**Contribution:** 2 fair
**Rating:** 6
**Confidence:** 2

**Summary:**

This paper introduces Pseudo-Likelihood Inference (PLI) as a novel method inside the simulation-based inference (SBI) family. SBI methods are used to estimate model parameters when the likelihood function is unknown or intractable. Existing SBI methods, such as Approximate Bayesian Computation (ABC) and Sequential Neural Posterior Estimation (SNPE), either approximate the likelihood or directly model the posterior. The contribution by the authors, PLI, combines neural approximation with ABC to achieve competitive performance in challenging Bayesian system identification tasks. The authors make use of integral probability metrics and introduce a smooth likelihood kernel with an adaptive bandwidth that is updated based on information-theoretic trust regions. This formulation allows for optimizing neural posteriors via gradient descent and does not rely on summary statistics. PLI performance improves the more data is available, particularly in scenarios involving stochastic simulations and multi-modal posterior landscapes. The effectiveness of PLI is evaluated through experiments conducted on four classical SBI benchmark tasks as well as a highly dynamic physical system. The results demonstrated particular advantages of PLI in handling stochastic simulations and multi-modal posterior landscapes.

**Strengths:**

* PLI is a novel approach addressing the limitations of existing methods that achieves competitive performance in challenging Bayesian system identification tasks. In fact, the proposed method outperforms SNPEs and other likelihood-free models, particularly when more data is available.

* The usage of the adaptive bandwidth for the smooth likelihood kernel allows for better adaptation to different problem settings and improves the accuracy of parameter estimation.

* The method enables to directly use gradient descent optimizing neural posteriors using gradient descent directly on raw observations or data points without requiring any intermediate summarization step.

* The effectiveness of PLI is evaluated in multiple experiments conducted on four classical Simulation-Based Inference benchmark tasks and in a highly dynamic physical system, demonstrating its applicability across various domains.

**Weaknesses:**

* The changes induced by replacing the KL divergence with the MMD and the Wasserstein distance should further explored. Consequences on changing the KL divergence for other metrics has been studied before in works such as [1], where an adversarial approach is used to evaluate the KL divergence and other divergences using only samples. I strongly suggest discussing this possibility in the article, since it seems that this alternative can be useful here.

* To provide more robust experimental results, when comparing different models I would suggest also using other proper scoring rules [2].

* Since efficiency is an important aspect when dealing with high-dimensional tasks, there is limited discussion about computational efficiency throughout the paper. Considering that likelihood-free inference methods can be computationally demanding, it would be valuable to discuss any potential limitations or considerations regarding computational resources required by PLI compared to other SBI approaches

* While the paper highlights PLI's advantages over SNPEs, it would be valuable to understand how PLI performs against other popular approaches s.a. Markov Chain Monte Carlo methods. A comparative analysis would provide a more comprehensive understanding of PLI's strengths and weaknesses relative to existing techniques.

* (_minor_) Due to the formulation of several function-space Bayesian inference models, if you deemed it fitting it could be interesting discussion on the function-space approaches such as [3-6]. I think most of these models can be encompassed as (S)NPE methods and that could enrich the discussion in the paper.

* (_minor_) Please correct typos, s.a. "posteriror" in line 143, "divergnce" in line 156.

* (_minor_) The red hyperlink boxes to equations appear with an extra line inside, which difficults reading. Please check this.

---

**References**:

[1] Santana, S. R., & Hernández-Lobato, D. (2022). Adversarial α-divergence minimization for Bayesian approximate inference. Neurocomputing, 471, 260-274.

[2] Gneiting, T., & Raftery, A. E. (2007). Strictly proper scoring rules, prediction, and estimation. Journal of the American statistical Association, 102(477), 359-378.

[3] Ma, C., & Hernández-Lobato, J. M. (2021). Functional variational inference based on stochastic process generators. Advances in Neural Information Processing Systems, 34, 21795-21807.

[4] Rodrı́guez Santana, S., Zaldivar, B., & Hernandez-Lobato, D. (2022, June). Function-space Inference with Sparse Implicit Processes. In International Conference on Machine Learning (pp. 18723-18740). PMLR.

[5] Ma, C., Li, Y., and Hernández-Lobato, J. M. (2019). “Variational implicit processes”. In: International Conference on Machine Learning, pp. 4222–4233.

[6] Sun, S., Zhang, G., Shi, J., and Grosse, R. (2019). “Functional variational Bayesian neural networks”. In: International Conference on Learning Representations.

**Questions:**

* What are the limitations and potential challenges associated with implementing PLI? The paper presents PLI as an effective method for challenging Bayesian system identification tasks, but it is important to address potential limitations or practical considerations when applying this approach in real-world scenarios. For example, what computational resources are required by PLI? Are there specific problem domains where its performance may be less favorable?

**Limitations:**

* The paper could improve by comparing PLI with more SBI methods.

* The evaluation focuses on a limited set of benchmark tasks and one physical system, potentially limiting the generalizability of PLI to diverse problem domains.

* There is insufficient discussion about the computational efficiency and resource requirements of implementing PLI.

---

> ### Author Rebuttal · Authors · 2023-08-09
>
> ### Answer to Reviewer AzJv
> We thank the reviewer for the valuable feedback and the relevant references that we will include in the final version of the paper.
>
> **Remarks and Questions:**
> > The changes induced by replacing the KL divergence with the MMD and the Wasserstein distance should further explored.
>
> Thank you for this suggestion. We provided further insights in our general rebuttal response above regarding the choice of the divergence measure. Indeed, we agree that the effect of the divergence measure deserves further study and that there are other choices that one could consider, e.g., Kernelized Stein Discrepancy (KSD), the adversarial estimation of the KL, or the alpha-divergences. We believe that it is a fruitful direction for future work, and we hope that our paper stimulates this research by showing strong empirical results using MMD and Wasserstein distances. We will include a paragraph in Section 3.3 that discusses adversarial approaches as a way to approximate the KL and provide a reference to the suggested paper:
> *Alternatively, one could exploit adversarial strategies to approximate the KL divergence, as explored in the studies by Mescheder et al. (2017) and Santana and Hernández-Lobato (2022). Incorporating these methods into the PLI framework requires training an additional adversarial model. We leave this investigation to future work.*
>
> > Using other proper scoring rules
>
> Thanks for this suggestion. Indeed, the choice of the evaluation metric is important, as was also noted in the SBI benchmark [1]. Currently, we employed MMD and Wasserstein distance, the former being an instance of a kernelized scoring rule and the latter having a clear geometric interpretation. Since both metrics yield similar results with growing $N$, we did not further investigate this choice. Are there any particular proper scoring rules you suggest we include in the final version of the paper?
>
> > Considering that likelihood-free inference methods can be computationally demanding, it would be valuable to discuss any potential limitations or considerations regarding computational resources required by PLI compared to other SBI approaches
>
> PLI has the same computational requirements as other SBI methods that train neural models, such as SNPE. Only compared to ABC does PLI take longer, but note that ABC does not have a training phase, whereas PLI can be queried very cheaply after it is trained. Please see our overall response for more details. We will add these clarifications to the final version of the paper.
>
> > While the paper highlights PLI's advantages over SNPEs, it would be valuable to understand how PLI performs against other popular approaches s.a. Markov Chain Monte Carlo methods.
>
> We have compared against SMC-ABC --- the strongest method amongst ABC-based approaches. MCMC-ABC is not so popular as it requires a fixed rejection threshold, making the problem significantly harder when the prior distribution is not close to the target distribution. We highlight that the reference posterior samples for the SLCP and SIR tasks are drawn from MCMC sampling with the true likelihood. Thus, we are already implicitly comparing against MCMC sampling and see that Wasserstein and MMD converge to the MCMC samples.
>
> > Due to the formulation of several function-space Bayesian inference models, if you deemed it fitting it could be interesting discussion on the function-space approaches
>
> As far as we know, Functional Variational Inference (FVI) has not been considered in the Likelihood-Free Inference (LFI) context so far. It sounds interesting and we will mention it as a potential future work, together with the references to the suggested papers.
>
> > What are the limitations and potential challenges associated with implementing PLI? Are there specific problem domains where its performance may be less favorable?
>
> We want to highlight two scenarios in which PLI's *"performance may be less favorable"*. First, PLI is not amortized and won't perform well when reference data of different parameter configurations is present. Second, PLI relies on the MMD or Wasserstein distance to assess the similarity of the data, which can be a hard task when only a few observations are available. Additionally, all SBI methods assume that the simulation matches the data-generating distribution closely. Accounting for modeling errors is an ongoing large field that is far from solved.
>
> [1] Lueckmann, J. M., Boelts, J., Greenberg, D., Goncalves, P., & Macke, J. (2021, March). Benchmarking simulation-based inference. In International conference on artificial intelligence and statistics (pp. 343-351). PMLR.

---

> > ### Comment · Reviewer_AzJv · 2023-08-12
> > **Brief response**
> >
> > First of all, I would like to thank the authors for addressing in detail the points raised in all reviews. I have read through it all and I must say I am quite satisfied by the responses overall, even more given all the updates and extra information included in the final version of the paper and the fact that the authors plan to release the code as well.
> >
> > I am inclined to maintain my rating (at least for the given moment), although I am in fact more favourable now to the acceptance of the paper than I was earlier.
> >
> > Thanks again for all the work!

---

> > > ### Author Response · Authors · 2023-08-14
> > >
> > > Thank you again for the time and investment in reviewing our proposed method. We are glad that the additional information helped clarify the remaining questions.

---

### Official Review · Reviewer_XuP4 · 2023-07-06

**Soundness:** 3 good
**Presentation:** 2 fair
**Contribution:** 3 good
**Rating:** 6
**Confidence:** 3

**Summary:**

This work aims at proposing a sequential Monte Carlo scheme to perform approximate Bayesian computation in the context of simulation based inference. Previous works focused on directly modeling posterior distributions whereas this approach presents a novel method to approximate likelihoods utilizing kernel density methods along with an information theoretic kernel bandwidth update scheme. The authors claim that this direction allows for improvement over previous works in the case of having more reference data to learn from.

**Strengths:**

I think that employing variational inference to help make SMC-ABC more tractable is an interesting direction. Further leveraging this setting to derive a trust-region compliant kernel bandwidth is significant and a good direction to explore. A number of different non-trivial steps are performed in order to derive the final pseudo-likelihood inference scheme and I believe this deserves commendation.

**Weaknesses:**

Almost all of the derivations are based on the classic variational inference objective of reducing the KL divergence between a true distribution and the variational distribution being learned; however, the actual method proposed substitutes this for alternative divergences/"integrated probability metrics" such as Wasserstein distance and MMD. I understand the necessity of this due to the likelihood being unavailable in SBI; however, this does undercut a lot of the impact that the various derivations have on the actual problem in my opinion.

I do not find the experimental results as convincing as they could be. For the first two benchmarks (Gaussian Location and Gaussian Mixture), it appears that MMD-ABC is either comparable to both versions this paper is proposing or better across all values of N and in both metrics. Additionally, in the appendix it can be seen in Figure C.1 that using sample based metrics (referred to as PPCs in the paper) the PLI methods appear to perform about the same or worse in several cases in all 5 tasks. I wouldn't go as far to say that they are always comparable or worse, but the story does not seem as clear cut as it is made out to be in the main set of results in Figure 3 in the main paper. Note that in line 285 it is justified to use PPCs instead of comparing posterior distributions, so I believe it should be valid to look at these results for the benchmarks as well.

Please feel free to correct me if I have misunderstood something as I could have easily missed some key details.

**Questions:**

Is there a clear explanation as for why the APT method starts to perform worse when given more reference data? Naively I would think that it would plateau in performance but it appears to severely degrade. We see somewhat similar behavior in some of the settings with both PLI and ABC methods, although to a much lesser degree (and more infrequent). I would be interested in learning if there is any insight into why this could be happening to all of these methods.

**Limitations:**

The authors did adequately discuss the limitations in my opinion.

---

> ### Author Rebuttal · Authors · 2023-08-09
>
> ### Answer to Reviewer XuP4
> We thank the reviewer for the valuable feedback regarding our proposed method.
>
> **Remarks and Questions:**
> > I understand the necessity of [introducing IPMs over the KL] due to the likelihood being unavailable in SBI; however, this does undercut a lot of the impact that the various derivations have on the actual problem in my opinion.
>
> It is indeed a crucial idea of our paper to use IPMs to compare samples when the KL divergence cannot be directly applied. Please see our detailed discussion on the impact of the different divergences in our general response.
>
> > For the first two benchmarks (Gaussian Location and Gaussian Mixture), it appears that MMD-ABC is either comparable to both versions this paper is proposing or better across all values of N and in both metrics.
>
> Please observe that the Gaussian Location and Gaussian Mixture tasks are simple benchmarks for which we can obtain the exact posterior and where the algorithms are expected to perform well. It is not surprising that ABC even slightly outperforms PLI on these tasks because it is not training a model of the posterior. On the more challenging tasks, such as APT and SIR, there is a significant drop in the performance of ABC while the PLI performance increases with increasing $N$. This is a key result of the paper since we want to use many observations instead of only one.
>
> > Additionally, in the appendix it can be seen in Figure C.1 that using sample based metrics (referred to as PPCs in the paper) the PLI methods appear to perform about the same or worse in several cases in all 5 tasks. Note that in line 285 it is justified to use PPCs instead of comparing posterior distributions, so I believe it should be valid to look at these results for the benchmarks as well.
>
> Please let us clarify how to interpret the additional PPC metrics in the appendix. Due to the intractability of the likelihood, the evaluation of SBI methods in the parameter space is in general infeasible. Therefore, we provided posterior predictive checks (PPCs) to evaluate the posterior predictive performance in the observation space  $\text{PPC}^D = \sum_i D(p^\star(x), p(x|\xi_i)),\,\xi_i \sim\pi(\xi_i)$. Your observation that the PPC scores are not matching the parameter space metrics is partially correct, although we restrict this observation to the SIR task. After submission, we found a small bug where the evaluation data was drawn from an incorrect ground truth parameter. We fixed the bug and reran the evaluations. The corrected results align with the paper's claims and show that PPC can be used as a valid performance measure. We checked all other experiments, and there the results are correct. Thank you for pointing out this inconsistency.
>
> > Is there a clear explanation as for why the APT method starts to perform worse when given more reference data?
>
> We hypothesize that it is due to the high dimensional observation space, as all observations are concatenated. As the simulation budget was fixed to make the methods comparable, this might explain the significant drop in performance. In our general rebuttal response above, we discuss the performance drop in further detail.
>
> > We see somewhat similar behavior in some of the settings with both PLI and ABC methods
>
> Here we would like to stress that this behavior is distinct from the behavior seen in APT. The slight decay of performance for large $N$, i.e., $N=1000$ can be attributed to the fact that the reference posterior samples are only approximations obtained from MCMC sampling for the SLCP and SIR tasks. For large $N$, obtaining reliable posterior samples with MCMC becomes a challenging task. Note that this phenomenon is not visible in the Gaussian location and Gaussian mixture tasks as these posteriors can be evaluated analytically.

---

> > ### Comment · Reviewer_XuP4 · 2023-08-14
> > **Reply to Authors**
> >
> > Thank you for the in depth reply to my questions and concerns. I still think the connections between the theorems derived and the actual proposal with the chosen divergence criteria is a bit tenuous, even with the added empirical results, and could benefit from some added analysis or slightly stronger ties; however, in light of my other points of interested being addressed I have decided to raise my score from a 5 to a 6.

---

### Author Rebuttal · Authors · 2023-08-09

### Answer to all Reviewers

We thank the reviewers for their insightful feedback. We are pleased that the **novelty and significance** of the paper are recognized by the reviewers.
- **XuP4:** A number of different non-trivial steps are performed [...] and I believe this deserves commendation
- **AzJv:** novel approach addressing the limitations of existing methods
- **9BA6:** methodological developments seem solid and well motivated

The reviewers point out multiple times that the **experiments support the paper’s claims**, that PLI is **competitive** compared to state-of-the-art baselines, and that the **extension to the novel Furuta pendulum** experiment is a strength:
- **AzJv:** The effectiveness [of PLI] is evaluated in multiple experiments [...], *demonstrating its applicability across various domains*.
- **9BA6:** Experiments seem to support the main message of the paper.
- **7dVy:** The Furuta pendulum, which I had not seen before, is particularly nice

We are further pleased about the positive comments regarding **presentation** and **clarity** of the paper:
- **9BA6:** the paper is well written and easy to follow
- **dHap:** The literature review is a strength and highlights the authors' knowledge on the topic.
- **7dVy:** method is fairly well presented

The reviewers raised several good questions. We want to answer these here:

**Choice of the discrepancy measure $D$.**
Some reviewers expressed interest in the theoretical justification for employing MMD or Wasserstein divergence in the pseudo-likelihood kernel. They also questioned whether Lemma 1's convergence result and Appendix A1's derivation for the KL are still valid in the context of other divergences. We aim to answer this question empirically by comparing our posterior to the reference posterior samples. We demonstrate that PLI posteriors, indeed, concentrate on the true posterior. Though rigorous theoretical convergence guarantees remain a work in progress, we cite MMD-Bayes [1] (l. 156-161) for concentration bounds around the posterior predictive distribution for a posterior based on an exponential MMD kernel. In contrast, there is no such result for W-PLI, despite its similarity to Wasserstein-ABC [2], which has convergence guarantees for the uniform kernel. We hope that our empirical findings will stimulate research into the concentration properties of PLI with Wasserstein and other divergences.

**Computational cost of PLI.**
As some reviewers asked about the computational cost, we will include the following summary in the final paper:

*In PLI, computational costs are spread across three components: simulation, summary statistic estimation, and normalizing flow training, all benefiting from GPU parallelization. In our experiments, neural model training accounted for the primary computational effort, while simulation and summary statistic computation were minor. PLI and SNPE required 60-90 min per task, compared to ABC's 2-10 min on an Nvidia RTX 3090. If a simpler model is used, e.g.,  GMMs, PLI's computation time matches ABC's. Commonly, high-fidelity simulators are employed in SBI methods, which are expensive to query. Therefore, simulations will likely constitute a substantial proportion of the total compute in more realistic situations.*

**Performance of APT drops for a large number of observations $N$.**
Several reviewers noticed that the performance of APT drops for large $N$. Since the increase in the number of reference observations means that the dimensionality of the input to the conditional normalizing flow (CNF) increases with $N$, the search space increases significantly for APT, which is trained on pairs of parameters and simulations. In contrast, PLI does not suffer from the increase of dimensionality because the density estimator is not conditioned on the data and has an additional inductive bias through the pseudo-likelihood evaluation.

**Updated experimental results on the SIR task.**
We investigated the issue raised by Reviewer XuP4 that the evaluation metrics in the observation space and the parameter space were disagreeing on the SIR task in Figure C.1. After the initial submission, we made the same observation and identified the source of that issue: the evaluation data (observations) for the SIR task had been generated from a different ground truth parameter yielding incorrect evaluation data. This bug explains the contradicting results between the evaluation in the observation space (upper two rows) against the metrics in the parameter space (lower three rows) in column 4. We have updated the evaluation data in the meanwhile and rerun the evaluation script for the SIR task (column 4). We further checked all other experiments, and there the data was correct. We provide an updated Figure C.1 in the rebuttal PDF.

**Choice of the evaluation tasks.**
Some reviewers suggested including more tasks from the SBI benchmark [3]. We want to clarify why we focused only on a subset of tasks. Namely, we did not use the same setup as in the SBI benchmark, where there is only $N=1$ observation, but we extended it to $N>1$ observations. Therefore, we aimed, on the one hand, at verifying our approach on tasks that have an analytical posterior (Gaussian Location Model and Gaussian Mixture Model) and on which all baseline methods are expected to work well, and on the other hand, at tackling the most challenging problems, SLCP and SIR, that require MCMC sampling to obtain the posterior samples on which the baselines would struggle. As getting accurate posterior samples with MCMC for increasing $N$ proved difficult, we restricted our evaluation to these four benchmark tasks.

[1] Chérief-Abdellatif & Alquier (2020). MMD-Bayes: Robust Bayesian estimation via maximum mean discrepancy.

[2] Bernton et al. (2019). Approximate Bayesian computation with the Wasserstein distance.

[3] Lueckmann et al. (2021). Benchmarking simulation-based inference.

---

### Comment · Area_Chair_ZTmw · 2023-08-15
**Author-reviewer discussion**

Dear all,

The author-reviewer discussion period has now started. It will continue for one more week, until August 21.

@authors: Please respond to the comments or questions reviewers may further have. Remain short and to the point.

@reviewers: Please read the author's responses and ask any further questions you may have. To facilitate the decision by the end of the process, please also acknowledge that you have read the responses and indicate whether you want to update your evaluation.

- You can update your evaluation positively (if you are satisfied with the responses) or negatively (if you are not satisfied with the responses or share other reviewers' concerns). Please note that major changes are a reason for rejection.
- You can also keep your evaluation unchanged. In this case, please indicate that you have read the responses and that you do not have any further comments.

Best regards,
The AC

---

### Decision · Program_Chairs · 2023-09-21

**Decision:**

Accept (poster)

**Comment:**

The reviewers all recommend acceptance (6-6-6-6-6). The author-reviewer discussion has led to a number of remarks, questions and suggestions that can be used to improve the paper. We request the authors to address as much as possible the reviewers' comments in the final version.